# Barriers for Learning in an Evolving World: Mathematical Understanding of Loss of Plasticity

**Amir Joudaki**[1], **Giulia Lanzillotta**[1], **Mohammad Samragh Razlighi**[2], **Iman Mirzadeh**[2]
**Keivan Alizadeh**[2], **Thomas Hofmann**[1], **Mehrdad Farajtabar**[2], **Fartash Faghri**[2]
[1]ETH Zürich   [2]Apple
amir.joudaki@inf.ethz.ch, {fartash, m_farajtabar}@apple.com

## Abstract

Deep learning models excel in stationary settings but suffer from loss of plasticity (LoP) in non-stationary environments. While prior literature characterizes LoP through symptoms like rank collapse of representations, it often lacks a mechanistic explanation for why gradient descent fails to recover from these states. This work presents a first-principles investigation grounded in dynamical systems theory, formally defining LoP not merely as a statistical degradation, but as an entrapment of gradient dynamics within invariant sub-manifolds of the parameter space. We identify two primary mechanisms that create these traps: frozen units from activation saturation and cloned-unit manifolds from representational redundancy. Crucially, our framework uncovers a fundamental tension: the very mechanisms that promote generalization in static settings, such as low-rank compression, actively steer the network into these LoP manifolds. We validate our theoretical analysis with numerical simulations and demonstrate how architectural interventions can destabilize these manifolds to restore plasticity.

**Code:** https://github.com/ajoudaki/loss-of-plasticity

## 1 Introduction

The extraordinary success of back-propagation in training deep neural networks often relies on two implicit assumptions. First, *stationarity* is assumed: the data distribution encountered during training is similar to the distribution faced during deployment, implying that post-training adaptation is minimal. Second, a *single random initialization* of network parameters is the main source of diversity and exploration potential—a resource that is progressively consumed by optimization and rarely replenished. These assumptions falter when an artificial agent must operate and learn continuously within an environment characterized by changing dynamics or evolving task distributions. This scenario, commonly referred to as continual or lifelong learning, presents the stability-plasticity dilemma (Abraham and Robins, 2005; Chaudhry et al., 2018), demanding a system stable enough to retain acquired knowledge yet plastic enough to integrate new information.

Empirically, standard deep networks subjected to long sequences of tasks or drifting data streams exhibit a sharp decline in learning capability (Dohare et al., 2024; Berariu et al., 2021; Nikishin et al., 2022). This phenomenon, termed *Loss of Plasticity* (LoP), is distinct from catastrophic forgetting: while forgetting erases old knowledge, LoP prevents the acquisition of new knowledge. Common symptoms are well-documented: exploding weight magnitudes (Nikishin et al., 2022), the emergence of "dead" units (Sokar et al., 2023; Lyle et al., 2022), and a collapse in the effective rank of representations (Papyan et al., 2020; Huh et al., 2023; Kumar et al., 2021). While recent works have linked LoP to back-propagation mechanics (Dohare et al., 2024) or NTK rank degradation (Lyle et al., 2023), these observations are largely descriptive. They characterize the *symptoms* of the failing network but stop short of a formal mechanistic explanation for the *persistence* of the non-plastic state.

A fundamental question remains: Why exactly does gradient descent fail to recover from these states? If LoP is merely a bad configuration, why do gradients not steer the model back toward useful regions? For example, while many works have identified low rank as a hallmark of LoP, why is the network unable to recover feature diversity under a new task distribution? Our work revisits

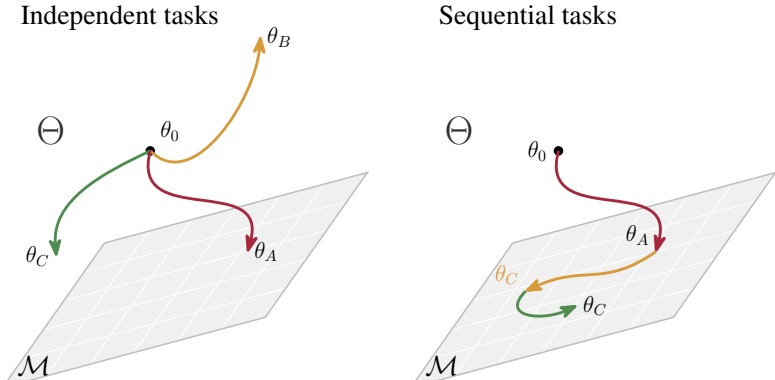

Figure 1.1: **Conceptual illustration of LoP as a topological trap.** The high-dimensional parameter space $\Theta$ contains low-dimensional, invariant *LoP manifolds* $\mathcal{M} \subset \Theta$ . **Left (Independent tasks):** During standard learning, trajectories starting at $\theta_0$ can explore the full space (e.g., $\theta_B, \theta_C$), though implicit biases may eventually drive them toward simpler representations on $\mathcal{M}$ ($\theta_A$). **Right (Sequential tasks):** In continual learning, early tasks drive parameters onto the manifold ($\theta_0 \to \theta_A$). Once trapped, gradients for subsequent tasks become strictly tangent to $\mathcal{M}$, constraining future dynamics to this restricted subspace ($\theta_A \to \theta_B \to \theta_C \to \dots$) and preventing the recovery of plasticity needed for new data distributions.

the dynamics of gradient descent through the lens of dynamical systems theory to answer this. We posit that LoP is not simply a degradation of statistics, like rank or weight norm, but a topological entrapment under gradient descent. We seek to identify the specific geometric structures underlying the optimization landscape that act as "sinks" for gradient trajectories, restricting the model from accessing its full degrees of freedom.

The central contribution of this work is the formalization of **LoP manifolds**: invariant sub-manifolds of the parameter space that act as "traps" for gradient descent (Fig. 1.1). Once the optimization process steers parameters into these regions, the gradient flow becomes strictly tangent to the manifold, rendering escape impossible without external intervention. This formulation fills a critical theoretical gap: while prior literature has extensively cataloged the *symptoms* of plasticity loss (e.g., weight explosion, rank collapse), our framework identifies the mathematical and geometric principles that provably make these states persistent and irreversible. Furthermore, we show the driver towards this entrapment—a fundamental **rank-plasticity tension**. We show that the very mechanisms that promote generalization in static settings, specifically, the compression of features toward low-rank structures, are the primary forces that steer the network into these LoP manifolds. Thus, the dynamics that maximize performance on the *current* task inadvertently construct the barrier to adaptability for *future* tasks.

**Summary of Contributions.** In this work, we formalize the mechanics of plasticity loss and validate our theory across varied architectures. Our specific contributions are:

- **A Dynamical Systems Definition of LoP (Sec. 2).** We move beyond symptomatic definitions to propose a formal definition of LoP as the entrapment of optimization trajectories within specific invariant sub-manifolds of the parameter space.

- **Identification of Trap Mechanisms (Sec. 2).** We theoretically characterize two primary classes of invariant manifolds: *Frozen-Unit Manifolds* ($\mathcal{M}_F$), caused by activation saturation, and *Cloned-Unit Manifolds* ($\mathcal{M}_C$), caused by representational redundancy. We prove that standard gradient-based optimization cannot escape these manifolds once entered.

- **The Rank-Plasticity Tension (Sec. 3).** We derive a theoretical connection between feature rank dynamics and plasticity. We show that properties widely considered beneficial for generalization (e.g., neural collapse) act as attractive forces toward LoP manifolds.

- **Empirical Validation and Mitigation (Sec. 4).** We validate our framework through extensive experiments on MLPs, ResNets, and ViTs. We further analyze how architectural interventions

(normalization) and targeted perturbations (noise injection, dropout) affect the stability of these manifolds, providing a principled basis for recovering plasticity.

## 1.1 BACKGROUND AND RELATED WORK

**Loss of Plasticity (LoP).** A network is said to suffer a loss of plasticity when, after some period of training, it can no longer acquire new information as effectively as a freshly-initialised model of the same architecture. LoP has been documented in a variety of continual-learning and reinforcement-learning settings (Dohare et al., 2024; Nikishin et al., 2022). Crucially, LoP is distinct from catastrophic forgetting: performance on past tasks may remain intact while the ability to learn future tasks degrades (Lyle et al., 2023). Typical symptoms include exploding weight norms, growing numbers of dead (saturated) units, and a collapse of the effective rank of hidden representations (Dohare et al., 2021; Papyan et al., 2020).

**Previous explanations.** Early accounts linked LoP to individual pathologies, e.g., weight-norm growth or activation sparsity. However, these factors alone failed to consistently explain the phenomenon (Lyle et al., 2022). A more recent view connects LoP to a degeneration of the network's neural tangent kernel (NTK) that is once the NTK becomes low-rank, many directions in function space receive negligible gradient and can no longer be learned (Lyle et al., 2023). This perspective suggests that LoP is multi-faceted with diverse surface-level defects (e.g., dead units, duplicated features) sharing the common consequence of reducing the network's effective degrees of freedom.

**Geometric Singularities and Learning Dynamics.** The connection between overparameterization, reduced dimensionality, and learning difficulties has deep roots in the analysis of neural network geometry. Hierarchical models exhibit singularities that are regions in parameter space where the mapping from parameters to function is not unique (e.g., due to unit duplication or vanishing). Foundational work by Fukumizu and Amari (2000) and Amari et al. (2006) demonstrated that these singularities cause the Fisher Information Matrix to degenerate, leading to slow learning dynamics (plateaus) as gradient descent is attracted to these regions.

**Implicit Bias and Stochastic Dynamics.** Recent work highlighted how the optimization algorithm itself contributes to the collapse towards simpler representations. Chen et al. (2023) analyzed the implicit bias of Stochastic Gradient Descent (SGD), showing that gradient noise induces an attractive force towards these singular regions (termed "Invariant Sets"). This "Stochastic Collapse" suggests that the tendency towards LoP states is exacerbated by the stochastic nature of the optimization process, even if the regions are unstable under deterministic gradient descent. Furthermore, Wang et al. (2025) empirically demonstrated that maintaining trainability (ability to fit new data) does not guarantee generalizability (performance on unseen data), emphasizing the need for methods that restore genuine plasticity.

## 2 LoP MANIFOLDS: TRAPS FOR GRADIENT DESCENT

In this section, we lay the groundwork for our analysis by defining Loss of Plasticity (LoP) within a dynamical systems framework and recalling the standard definitions for feed-forward neural networks and back-propagation. The stability of LoP manifolds, a crucial concept for understanding their persistence, will be discussed later in Sec. 2. Let $\theta \in \Theta \subseteq \mathbb{R}^p$ represent the parameters of a neural network. We consider training on a stream of data $\{(x_i, y_i)\}_{i=1}^{N}$ using gradient descent or its stochastic variants. The objective is typically to minimize a loss $\sum_{i=1}^{N} \mathcal{L}(\hat{y}_\theta(x_i), y_i)$, which we can succinctly refer to as $\mathcal{L}(\theta)$. This allows us to define LoP based on the trajectory of parameters $\theta(t)$ in the parameter space $\Theta$ as driven by the negative gradient in the loss landscape. As illustrated in the right panel of Fig. 1.1, we are interested in scenarios where this trajectory becomes confined to a lower-dimensional subspace $\mathcal{M} \subset \Theta$.

**Definition 2.1** (LoP Manifold). *A manifold $\mathcal{M} \subset \Theta$ induces LoP if the gradient of the loss function is tangent to the manifold at every point on the manifold. That is, $\nabla_\theta \mathcal{L}(\theta) \in T_\theta \mathcal{M}$ for all $\theta \in \mathcal{M}$, where $T_\theta \mathcal{M}$ denotes the tangent space of $\mathcal{M}$ at $\theta$. This tangency condition ensures that once the gradient flow enters $\mathcal{M}$, it remains within $\mathcal{M}$ under the dynamics of gradient flow $\frac{d\theta(t)}{dt} = -\nabla_\theta \mathcal{L}(\theta(t))$.*

**Remark 2.1.** *Definition 2.1 is stated in terms of continuous-time gradient flow $\dot{\theta} = -\nabla_\theta \mathcal{L}(\theta)$: the tangency condition $\nabla_\theta \mathcal{L}(\theta) \in T_\theta \mathcal{M}$ ensures that the flow starting in $\mathcal{M}$ remains in $\mathcal{M}$. For a general*

*curved manifold this need not imply invariance under discrete updates $\theta_{k+1} = \theta_k - \eta_k \nabla_\theta \mathcal{L}(\theta_k)$, as discretization step can lead to an escape from the manifold. However, for* affine *LoP manifolds, both gradient flow and (stochastic) gradient descent dynamics are invariant once they enter $\mathcal{M}$, as the discretization step cannot lead to an escape.*

**Remark 2.2.** *If the conditions in Definition 2.1 hold irrespective of the specific data distribution generating the loss $\mathcal{L}$, which we can think of as functional LoP, and is our primary area of interest. Such LoP arises from the network architecture and gradient descent dynamics alone and is particularly relevant as it persists even if the task or data distribution evolves.*

Given these definitions, we can formalize existence of these LoP manifolds, restricting subsequent learning. We present a central theorem that jointly addresses LoP arising from frozen and duplicate units. The intuition is that once units become unresponsive (frozen) or perfectly redundant (cloned), they tend to remain so under standard gradient-based optimization.

**Theorem 2.1.** *Let $G = (V, E)$ be the network's computational DAG and let $\theta = \{\theta_{uv} : (u \to v) \in E\} \in \Theta$ denote the edge parameters.*

1. ***Frozen-unit manifold $\mathcal{M}_F$.*** *Assume there exists $F \subset V$ such that, for all finite inputs encountered, each $v \in F$ is persistently saturated ($f'(z_v) = 0$). Then the gradients w.r.t. all incoming parameters to $v$ vanish on any mini-batch, so those coordinates remain fixed; writing the linear constraints as $\theta_{\text{in}}(v) = \text{const}$ for all $v \in F$, the affine subspace $\mathcal{M}_F := \{\theta : \theta_{\text{in}}(v) = \text{const} \ \forall v \in F\}$ satisfies $\nabla \mathcal{L}(\theta) \in T_\theta \mathcal{M}_F$ and GD/SGD updates initialized in $\mathcal{M}_F$ remain in $\mathcal{M}_F$.*

2. ***Cloning manifold $\mathcal{M}_C$.*** *Assume a partitioning of nodes into disjoint blocks $\{S_1, \ldots, S_k\}$ exists with following properties. For every ordered block pair $(S_i, S_j)$, we have the linear equalities $\sum_{v \in S_j} \theta_{uv} = \sum_{v \in S_j} \theta_{u'v}$ for all $u, u' \in S_i$ (equal row-sums) and $\sum_{u \in S_i} \theta_{uv} = \sum_{u \in S_i} \theta_{uv'}$ for all $v, v' \in S_j$ (equal column-sums). Let $\mathcal{M}_C$ be the affine subspace of $\Theta$ consisting of all $\theta$ satisfying these constraints. If $\theta \in \mathcal{M}_C$, then (i) all units within any block share the same forward values on any input, (ii) all units within any block share the same backpropagated errors on any input, and therefore (iii) the per-edge gradients are constant across edges connecting the same block pair, i.e., for any $(u, v)$ and $(u', v')$ with $u, u' \in S_i$ and $v, v' \in S_j$, $\partial \mathcal{L}/\partial \theta_{uv} = \partial \mathcal{L}/\partial \theta_{u'v'}$. Hence $\nabla \mathcal{L}(\theta) \in T_\theta \mathcal{M}_C$ and GD/SGD updates initialized in $\mathcal{M}_C$ remain in $\mathcal{M}_C$.*

Note that both LoP manifolds $\mathcal{M}_F$ and $\mathcal{M}_C$ are defined as linear LoP manifolds in the sense of Definition 2.1. Formal proofs and further details are provided in Appx. A.3.

**Proof idea.** *Frozen units.* If a unit stays in a regime with $f'(z_v) = 0$ for all finite inputs (e.g., $\tanh$ with very large $\|\theta_{\text{in}}(v)\|$ or ReLU with a large negative bias), then $\partial \mathcal{L}/\partial \theta_{\text{in}}(v) \approx 0$; its incoming parameters are fixed, so updates are tangent to $\mathcal{M}_F$. *Cloning via redistribution.* The key idea the row/column-sum equalities mean total incoming/outgoing weight from/to any block is redistributed within each block pair $(S_i, S_j)$. Thus, the total contribution to the forward and backward of each unit within a block remains identical, implying the forward and backward cloning (properties (i) and (ii)) within blocks. Thus, per-edge gradients $d\mathcal{L}/d\theta_{uv} = h(u)\,\delta(v)$, are therefore by forward and backward symmetries across the blocks the gradients will be constant for any two units in these blocks $(u, v) \in S_i \times S_j$. These block-wise constant gradients trivially satisfy the row-sum and column-sum equalities, and hence are tangent to $\mathcal{M}_C$, and first-order updates remain on both manifolds.

**Remark 2.3.** *It is important to note that the Duplicate Manifold $\mathcal{M}_D$ (defined formally via Incoming and Outgoing Equitable partitions in Appx. A.3) represents a significant generalization of the cloning concepts typically discussed in literature. Prior analyses of singularities (Fukumizu and Amari, 2000) or invariant sets (Chen et al., 2023) generally define cloned units by requiring their associated weights to be strictly identical (the block-wise constant condition in our terminology). Our framework proves that invariance under gradient descent holds even under the relaxed condition of equitability, where individual weights may differ as long as specific incoming and outgoing sums are maintained. This significantly broadens the class of structures identified as LoP manifolds.*

**Remark 2.4.** *The cloning LoP manifold naturally lends itself to gradient descent and stochastic gradient descent, regardless of the order which we process the samples, will remain strictly within the manifold. This extends to virtually all variations of gradient descent based optimizations, namely Stochastic Gradient Descent (SGD), SGD with momentum, and Adam, as long as the optimizer is*

*initialized at the onset of cloning. The only exception to this is weight decay which could break some symmetries. This fact can be empirically observed across our cloning experiments, showing that across a wide range of optimization schemes the model remains trapped onto to the LoP manifold.*

Remarkably, the theorem admits a *modular* version, which allows us to create practical cloning certificate for modern architectures (see Appx. A.4 for more details).

**Theorem 2.2** (Modular Cloning (informal)). *This cloning property can be decomposed modularly. If a network is composed of individual modules (e.g., layers or blocks), and each module locally satisfies the cloning invariance properties—namely, (1) cloned inputs produce cloned outputs (Forward Invariance), (2) cloned backward signals at the outputs produce cloned backward signals at the inputs (Backward Invariance), and (3) gradient updates preserve these invariances (Persistence)—then the entire network resides on a cloning manifold, provided the cloning profiles (partitions) are consistent at the interfaces between modules.*

To empirically test the validity of the cloning manifold and their potential escape mechanisms we conduct *cloning experiments*. First, a base model (e.g., an MLP) is trained on a specific task. Subsequently, a larger model is constructed by expanding the base model. This expansion involves increasing the width of the model for MLPs, the number of channels for CNNs and ResNets, and the feature dimension for ViTs. The weights of the cloned model are initialized in such a way that its activations are identical to those of the base model. This effectively creates blocks of units that have identical activations. Next, we train both the base and cloned models on the same task and monitor their training progress through the loss curve, the effective rank of representations, and the cloning $R^2$ score. Figure 2.1 presents the results of such experiments on MLPs, shedding light on the dynamics within and escapes from these LoP manifolds. The empirical validation of these claims, such as demonstrating perfect cloning under specific initializations or the persistence of dead units, can be found in Fig. 2.1 and Appx. B. Notably, Remark 2.4 predicts that Adam, despite fundamental differences with SGD, also fails to escape the manifold, which is validated by our experiments.

**Escaping the LoP manifolds with perturbations.** While a comprehensive theoretical analysis of the stability of empirically observed LoP manifolds is beyond the scope of this work, our empirical investigations indicate that these manifolds are frequently unstable or resemble saddle-like shapes. Certain types of noise or symmetry-breaking operations can help models escape these manifolds. We highlight two common perturbations: (1) *Noisy SGD* is a modification of SGD that adds Gaussian noise to the computed gradients before parameter updates. The magnitude of this injected noise is usually proportional to the norm of the gradient, with its initial relative strength gradually decreasing over successive steps. By applying this noise after cloning, we can determine whether the model can escape the LoP manifold or if it will fall back. (2) *Dropout* introduces stochasticity in the forward and backward passes by randomly zeroing activations. For cloned units, this breaks the symmetry because different clones might be active in different dropout masks, leading to divergent gradient updates. This is supported by experiments where dropout helps a model escape an artificially induced cloning manifold (see Fig. 2.1).

Both noisy SGD and dropout act as symmetry-breaking operations. In the case of dropout, both forward and backward passes are asymmetric for cloned units. For noisy SGD, the backward pass (gradient update) becomes asymmetric. This asymmetry causes the parameters of notionally cloned units to slowly diverge. Remarkably, in our MLP experiments, even a small amount of gradient noise, e.g., a single step with noise magnitude $0.01$ relative to gradient norms, suffices to initiate escape from an LoP manifold, though stronger noise generally leads to faster escape. In contrast, in settings such as Vision Transformers, while the model could escape from the manifold with a small perturbation, it did not move very far from it. More experimental studies into this direction would be vital to better understand the stability of these LoP manifolds.

## 3 EMERGENCE OF LOSS OF PLASTICITY FROM LINEAR–NONLINEAR RANK DYNAMICS

Having established the existence of LoP manifolds, we now discuss the mechanisms within standard training that drive their formation. The optimization process naturally follows a trajectory beginning with an expansion of representational diversity, as features propagate through nonlinear layers and under some configurations can become increasingly decorrelated (Poole et al., 2016). More recently, it

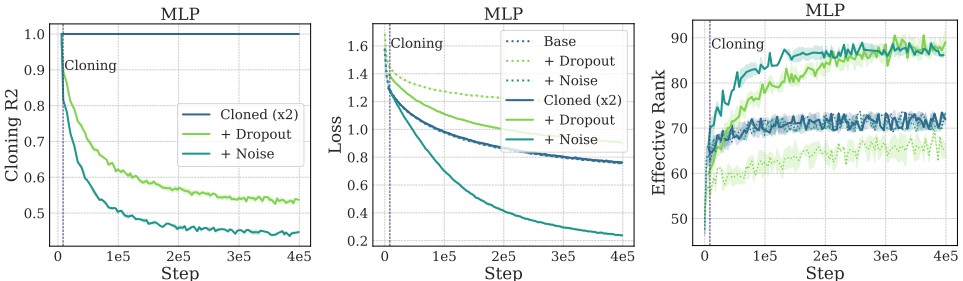

Figure 2.1: Cloning MLPs experiments. The empirical data validates Theorem 2.1 on duplicate manifold LoP. The cloned network dynamics remain confined in the base network manifold when using SGD, however using Noisy SGD or Dropout the dynamics can escape the manifold. **Left**: Cloning $R^2$ score quantifies the proportion of variance in individual unit activations within a cloned block that is explained by the mean activation of that block. An $R^2$ score of 1 indicates perfect cloning (units in a block are nearly identical), while a 0 score indicates no explained variance. See Appx. B.1.2 (Appendix B) for the precise formula and calculation details. **Middle**: Training loss comparison. *Cloned loss* refers to the loss of the cloned model during its training phase, while *base loss* refers to the loss of the original base model, which continues training for comparison. **Right**: Effective rank evolution showing representational diversity.

was shown that the layerwise kernel map has a globally attracting fixed-point structure that determines whether representations become orthogonal, partially aligned, or fully aligned based on the activation and hyper-parameters (Joudaki and Hofmann, 2025). This orthogonalization has also been attributed to batch normalization (Daneshmand et al., 2021). However, this expansion is counterbalanced by a drive toward geometric simplicity in the final phase of training, as captured by neural collapse (Papyan et al., 2020), where within-class variability vanishes and features converge to a low-rank subspace spanned by class means. We argue that this geometric collapse—while beneficial for generalization in static settings—inherently steers the model toward the low-rank LoP manifolds we identified. Thus, the very dynamics that maximize separability on the current task provide the direct mechanism for the emergence of LoP.

To diagnose whether features are diversifying or compressing during training, we track a smooth surrogate of the rank of the feature correlation matrix. Exact rank is numerically unstable, because it is not a continuous nor a differentiable map from the matrix space. Therefore, we use differentiable proxies to rank such as Rényi-2 rank, $\mathrm{er}_2(M) = (\mathrm{tr}\, M)^2/\|M\|_F^2$, or the Shannon effective rank, $\mathrm{er}(M) = \exp(H(\lambda(M)/\mathrm{tr}\, M))$. Both increase when the eigenmass is evenly distributed and decrease when dominated by a few values. Note that both of these surrogates are maximized when matrix $M$ has equal eivenvalues and minimized when it a rank-1 matrix. The following theorem offers an insight into how nonlinear layers contribute to the formation of features.

**Theorem 3.1** (Rank gain and decorrelation potential). *Assume $\phi \in L^2(\mathcal{N}(0,1))$ is nonlinear and normalized such that $\mathbb{E}[\phi(Z)] = 0$ and $\mathrm{Var}(\phi(Z)) = 1$. Let $C$ be a pre-activation correlation matrix. For an activation $\phi$, define the correlation kernel $K_\phi(r) = \mathrm{Corr}(\phi(x), \phi(y))$ where $(x, y)$ are jointly Gaussian with correlation $r$. The nonlinearity acts entrywise on correlations, producing $K_\phi(C)$. We define the* decorrelation strength *$\gamma_\phi = \frac{1 - K'_\phi(0)}{1 + K'_\phi(0)}$ and the* decorrelation potential *$\Psi(C) = \sum_{i \neq j} C_{ij}^2 (1 - C_{ij}^2)$. The Rényi-2 effective rank is non-decreasing according to the bound below:*

$$\frac{\mathrm{er}_2(K_\phi(C))}{\mathrm{er}_2(C)} \geq 1 + \gamma_\phi \frac{\Psi(C)}{\|C\|_F^2}.$$

*The ratio is strictly larger than 1 when the potential $\Psi(C)$ is vanishing (all off-diagonal $C_{ij} \in \{0, \pm 1\}$) and the the activation is linear ($\gamma_\phi$ is zero).*

Note that the theorem implies that any correlations within range $|C_{ij}| \in (0, 1)$ provide a potential $\Psi(C) > 0$ that the nonlinearity consumes to increase rank. The rate of this increase is governed by the scalar $\gamma_\phi \in [0, 1]$, depends on the activation and quantifies the nonlinearity's ability to de-correlate inputs and thereby increase rank. Larger $\gamma_\phi$ guarantees a larger rank gain for the same input spectrum. For a rigorous derivation, see Appx. A.1.

**Implication for emergence of frozen units.** In practice, the first and second moments of pre-activations drift during training, modulating the effective activation function. This modulation directly impacts the decorrelation strength $\gamma_\phi$. We find that $\gamma_\phi$ is *monotonic* with respect to the saturation parameters: for ReLU, making the effective bias more negative increases $\gamma_\phi$; for tanh, increasing the effective gain does the same. Consequently, optimizing for feature diversity (high decorrelation strength) creates a pressure to push units into regimes where the derivative vanishes on most inputs. This explains why training that effectively restores rank also creates "frozen" or "dead" units.

**Implication for creation of duplicate or cloned units.** Neural collapse is a widely observed endpoint in which the penultimate features are low-rank, and the class means form a simplex or Equiangular Tight Frame (ETF) structure (Papyan et al., 2020). The theorem makes a precise prediction regarding this state: for the rank to stabilize (ratio $\approx 1$) while retaining nonlinearity ($\gamma_\phi > 0$), the decorrelation potential $\Psi(C)$ must vanish. This forces correlations to converge to the fixed points 0 (orthogonal features) or $\pm 1$ (aligned or anti-aligned features). Thus, the dynamics of rank expansion naturally drive the network toward a mixture of orthogonal subspaces and cloned units, matching the geometry of LoP manifolds.

The two implications above provide a theoretical perspective on why duplicate features and frozen units frequently emerge at or near convergence, and why the resulting representation lies close to LoP manifolds, as proven in Theorem 2.1.

**Remark 3.1.** *Our analysis explicitly links LoP to the "Rich" or feature learning regime. In the "Lazy" learning regime (associated with the Neural Tangent Kernel), weights remain close to initialization, maintaining high-rank connectivity and avoiding feature compression. Conversely, the "Rich" regime is characterized by significant weight evolution and the emergence of low-rank structures (Chizat et al., 2019; Woodworth et al., 2020). The frozen units and cloned manifolds we identify are essentially symptoms of this feature learning process carried to an extreme. Thus, LoP can be viewed as a pathology specific to the Rich regime: the same mechanism that enables deep networks to learn efficient representations (data compression) eventually traps them in low-rank manifolds that hinder future adaptability.*

### 3.1 EXPERIMENTAL VALIDATION

We validate our theory with experiments on MLP, CNN, ResNet, and ViT architectures, training them continually on a sequence of 40 5-class tasks derived from Tiny ImageNet. We track the emergence of LoP symptoms, including dead units, duplicate units, and effective rank degradation. Full experimental details are provided in Appx. B. The experimental evidence confirms our intuitions (see Appx. B and Figs. 3.1, 3.2, B.4 and B.6): depending on the architecture, we observe that a degradation in the model performance is concomitant with the emergence of duplicate or frozen units, and a corresponding decrese in representational diversity. Our inquiry so far highlights two

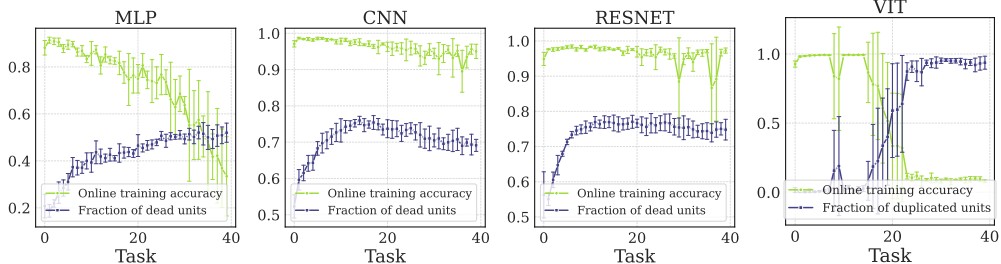

Figure 3.1: Causes and symptoms of Loss of Plasticity emerging during continual learning. The plots illustrate (across different architectures like MLP, CNN, ResNet, and ViT from left to right) an increase in the fraction of dead or duplicate units during training, coincidental with a decrease in training accuracy. These are key indicators of LoP. (Details of experimental setup in Appx. B).

key pathways to LoP common symptoms: (1) Emergence of **duplicate features**, where distinct computational units, or groups of units, within a network layer effectively learn to become identical or highly correlated, as a potential consequence of attempting to lower representational rank, (2) Emergence of **frozen or dead features**, where weights and biases of a unit stop learning, as a result

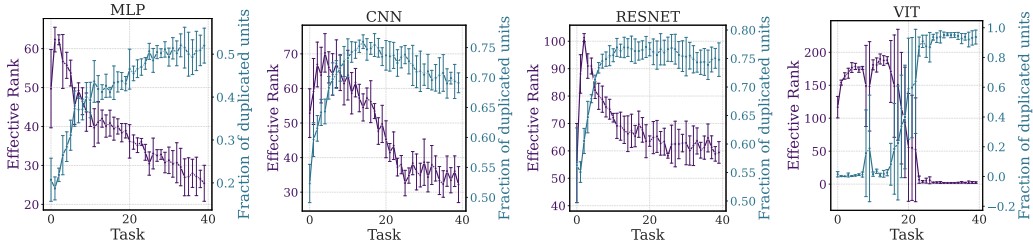

Figure 3.2: Co-evolution of Effective rank and LoP symptoms, such as dead or duplicate units in the network during continual training. (Experimental details in Appx. B).

of attempting to maximize rank increase (leading to saturation) or to flatten the loss landscape around the current parameters.

# 4 MITIGATION AND RECOVERY STRATEGIES

Having discussed the emergence of LoP symtoms and the existence of LoP manifolds, we now turn to strategies for preventing their formation or recovering from them if they have already occurred.

**Preventing LoP with Normalization.** As established in Sec. 3, one primary cause for activations becoming frozen is their pre-activations drifting into saturated regions. It is therefore natural to expect that normalization layers like Batch Normalization (BN) or Layer Normalization (LN) can help prevent this. By standardizing pre-activation statistics, these layers can keep activations operating in their more dynamic, non-linear range. Even with learnable affine parameters $(\gamma, \beta)$ after normalization, these parameters often act to maintain pre-activations within a "healthy" range, rather than pushing them into extreme values that cause saturation (e.g., consistently negative for ReLU). This is widely supported by empirical evidence (see Appx. B, Figures like Fig. 3.1 in Sec. 3, and Fig. B.3 ). BN and LN generally help maintain higher effective rank of representations throughout training (as seen in Fig. 3.2) and concurrently prevent frozen/dead features and excessive feature duplication from becoming dominant. This is consistent with theory showing that normalization layers bias the representation Gram matrix toward isometry (the identity) — a bias shown to persist during and after training, and to counteract the rank collapse associated with LoP (Joudaki et al., 2023).

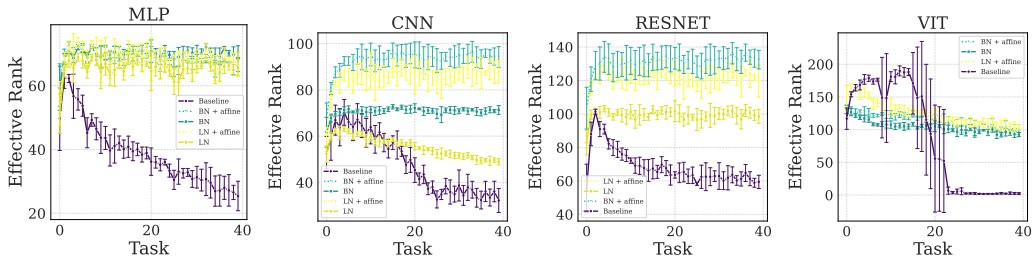

Figure 4.1: Evolution of the Effective rank during training for architectures with and without normalization layers. Dotted lines represent normalization with affine parameters. (Experimental details in Appx. B).

**Recovery from LoP via Perturbations.** What if LoP conditions, such as widespread frozen units or extensive feature cloning, have already set in? In such cases, mitigation strategies like normalization, which act proactively, may no longer be sufficient to reverse the state, as indicated by cloning experiments where normalization alone doesn't break perfect, established clones. However, similar to our discussion on manifold stability (Section 2), injecting noise into the training process can be a viable recovery strategy. The principle is that if the LoP manifold is unstable or saddle-like, perturbations can allow the optimizer to find an escape route. Noisy SGD and the more sophisticated Continual Backpropagation (Dohare et al., 2024) are examples of such mechanisms. We test recovery from LoP on the "bit-flipping" benchmark, an online regression task with a non-stationary target function designed to challenge a model's adaptability. A detailed description of

the task is in Appx. B.1.1. In order to demonstrate the recovery potential of noise injection, we design an experiment where the first half of 5M samples is processed by plain Stochastic Gradient Descent (SGD) and in the second half we switch the learning rule to Continual Backpropagation (CBP). Figure B.1 clearly shows a reversal in trend when the switch happens: whereas SGD causes the representations' rank to drop and the online training loss to increase, CBP amplifies the features' rank and reduces the online training loss, effectively *recovering plasticity*. Additionally Fig. B.2 illustrates how aspects like rank and feature duplication are affected by the dimensionality of the model. The disparity between SGD and CBP is only increased by the model size, hinting that the model scale might aggravate the symptoms of LoP. For details see Appx. B.1.1.

An interesting distinction arises when comparing artificially induced LoP (like explicit cloning) with naturally emerging LoP symptoms in challenging scenarios like continual learning. In controlled cloning setups (e.g., as conceptualized in Fig. 2.1, dropout can be effective in breaking the artificially imposed symmetry and allowing units to diverge. In contrast, in our continual Bit Flipping experiments, the role of dropout can be mixed or even detrimental. While it might prevent some forms of LoP, it can also hinder the consolidation of new knowledge or exacerbate forgetting if it too aggressively discards learned information relevant to the new task. This suggests that the optimal strategy for maintaining or recovering plasticity might be context-dependent.

## 5 CONCLUSION

This work presented a dynamical-systems framework for Loss of Plasticity (LoP) in deep neural networks. We formally defined LoP manifolds as regions in parameter space that trap gradient-based optimization, and identified two mechanisms for their formation: activation saturation leading to frozen units, and representational redundancy manifesting as cloned-unit manifolds. These LoP states are frequently characterized by reduced effective rank of representations. We investigated how normalization can mitigate LoP, and how perturbations like noise injection can facilitate escape, depending on manifold stability.

A key finding is the inherent tension between learning objectives in static and dynamic environments: properties that aid generalization on a fixed dataset, such as low-rank features or simplicity biases, can cause a loss of adaptability when learning extends over time or across changing tasks. This suggests continual learning needs mechanisms that actively preserve or regenerate representational diversity.

Several questions remain open. Theoretically, our analysis focused on linear or affine LoP manifolds; whether non-linear LoP manifolds exist and arise in practical training remains open. A fuller understanding of the stability conditions for different LoP manifolds—and the architectural or data conditions favoring one type over another—is also needed.

Numerically, the curvature normal to an LoP manifold is critical: for an unstable manifold with near-flat negative curvature, escape may require large perturbations or many training steps. Characterizing these curvatures and their impact on escape dynamics would be valuable. While we have demonstrated that models can escape artificially cloned LoP manifolds with interventions like dropout or noise, the question remains: can a model, once recovered from such a state, explore the parameter space as effectively and find solutions as generalizable as a model trained from a fresh, random initialization? This is of practical importance: whether exploratory capacity can be fully restored after collapsing into a highly restricted subspace.

An intriguing outcome of this work is the connection between unit cloning—often studied in model compression or network analysis—and LoP in continual learning, which have largely been treated as separate fields. Our framework, particularly the theorems on cloned units, reveals a deep link, suggesting that insights and tools can transfer between these domains. This raises whether continual-learning techniques such as noisy backpropagation or Continual Backpropagation (CBP) (Dohare et al., 2024) could help model expansion or escaping cloned states in other scenarios.

Ultimately, understanding and overcoming LoP is crucial for AI systems that learn continuously and adapt robustly in an ever-changing world. By providing a mathematical characterization of fundamental barriers to such adaptation, we aim to enable new architectures and learning algorithms that sustain plasticity indefinitely, leading to truly lifelong learning agents.

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

# A  THEORETICAL APPENDIX

This section contains detailed proofs of theorems and lemmas, further theoretical derivations, and discussions extending the concepts presented in the main paper.

## A.1  FORMAL PROOF OF THE RANK GAIN AND DECORRELATION THEOREM

This section provides the rigorous derivation of Theorem 3.1, establishing a quantitative lower bound on the increase in Rényi-2 effective rank across a nonlinear layer.

**Preliminaries: Hermite Expansion and Kernel Structure**  We consider an activation function $\phi \in L^2(\gamma)$, where $\gamma = \mathcal{N}(0, 1)$. Let $\{h_k\}_{k \geq 0}$ be the orthonormal Hermite basis. The activation admits the expansion:

$$\phi(z) = \sum_{k=0}^{\infty} a_k h_k(z), \qquad a_k = \mathbb{E}\big[\phi(Z)h_k(Z)\big].$$

We follow the assumptions of Theorem 3.1, where $\phi$ is standardized such that $\mathbb{E}[\phi(Z)] = 0$ (implying $a_0 = 0$) and $\mathrm{Var}(\phi(Z)) = 1$ (implying $\sum_{k \geq 1} a_k^2 = 1$).

The correlation kernel $K_\phi(r)$ is defined for jointly Gaussian $(X, Y)$ with correlation $r$:

$$K_\phi(r) = \mathrm{Corr}\big(\phi(X), \phi(Y)\big).$$

Using Mehler's identity, the kernel is given by:

$$K_\phi(r) = \sum_{k=1}^{\infty} a_k^2 r^k = \sum_{k=1}^{\infty} w_k r^k, \tag{1}$$

where $w_k = a_k^2$. Key properties derived from this structure are:

1. $w_k \geq 0$ for all $k$.
2. $\sum_{k=1}^{\infty} w_k = 1$, thus $K_\phi(1) = 1$ and $K_\phi(0) = 0$.
3. The linearity coefficient is $\alpha_\phi = K_\phi'(0) = w_1$. If $\phi$ is nonlinear, $\alpha_\phi < 1$.
4. $|K_\phi(r)| \leq |r|$ for $r \in [-1, 1]$.

This Hermite expansion of the activation correlation kernel $K_\phi$ and the analysis of its fixed points follow the framework laid out by Joudaki and Hofmann (2025) (there applied to the sample kernel; here to the feature–feature correlation matrix under the Gaussian pre-activation assumption).

**The Universal Quartic Gap Bound**  The key to the quantitative rank gain theorem is establishing a rigorous lower bound on the decorrelation gap $G(r) = r^2 - K_\phi(r)^2$.

**Lemma A.1** (Universal Quartic Gap Bound). *Let $\alpha = K_\phi'(0)$. The decorrelation strength is defined as $\gamma_\phi = \frac{1-\alpha}{1+\alpha}$. For any correlation $r \in [-1, 1]$:*

$$r^2 - K_\phi(r)^2 \geq \gamma_\phi \cdot r^2(1 - r^2). \tag{2}$$

*Proof.* We aim to find the kernel that maximizes $|K_\phi(r)|$ for a fixed $\alpha$, subject to the constraints $w_k \geq 0$ and $\sum w_k = 1$.

Let $x = |r| \in [0, 1]$. Due to the non-negativity of $w_k$:

$$|K_\phi(r)| = |\sum_{k=1}^{\infty} w_k r^k| \leq \sum_{k=1}^{\infty} w_k |r|^k = K_\phi(x).$$

We now seek an upper bound for $K_\phi(x)$. Since $x \in [0,1]$, we have $x^k \leq x^2$ for all $k \geq 2$.

$$K_\phi(x) = w_1 x + \sum_{k=2}^{\infty} w_k x^k$$

$$\leq \alpha x + x^2 \sum_{k=2}^{\infty} w_k$$

$$= \alpha x + x^2 (1 - \alpha).$$

Let $Q(x) = \alpha x + (1 - \alpha)x^2$. This quadratic function is the extremal kernel that maximizes the magnitude for a given $\alpha$ under these constraints.

Now we bound the gap $G(r)$:

$$G(r) = r^2 - K_\phi(r)^2 \geq x^2 - Q(x)^2.$$

We must verify that $x^2 - Q(x)^2 \geq \gamma_\phi x^2(1 - x^2)$.

$$x^2 - [\alpha x + (1 - \alpha)x^2]^2 \geq \frac{1 - \alpha}{1 + \alpha} x^2 (1 - x^2).$$

For $x \in (0, 1)$, we divide both sides by the positive term $x^2(1 - x)$:

$$\frac{1 - [\alpha + (1 - \alpha)x]^2}{1 - x} \geq \frac{1 - \alpha}{1 + \alpha}(1 + x).$$

We factor the LHS numerator as a difference of squares $1 - Y^2 = (1 - Y)(1 + Y)$, where $Y = \alpha + (1 - \alpha)x$. Note that $1 - Y = 1 - \alpha - (1 - \alpha)x = (1 - \alpha)(1 - x)$. The LHS simplifies to:

$$\text{LHS} = \frac{(1 - \alpha)(1 - x)[1 + \alpha + (1 - \alpha)x]}{1 - x} = (1 - \alpha)[1 + \alpha + x(1 - \alpha)].$$

The inequality becomes:

$$(1 - \alpha)[1 + \alpha + x(1 - \alpha)] \geq \frac{1 - \alpha}{1 + \alpha}(1 + x).$$

Assuming $\phi$ is nonlinear ($\alpha < 1$), we can divide by $(1 - \alpha)$. We then multiply by $(1 + \alpha)$:

$$(1 + \alpha)[1 + \alpha + x(1 - \alpha)] \geq 1 + x.$$

$$(1 + \alpha)^2 + x(1 - \alpha)(1 + \alpha) \geq 1 + x.$$

$$1 + 2\alpha + \alpha^2 + x(1 - \alpha^2) \geq 1 + x.$$

$$1 + 2\alpha + \alpha^2 + x - \alpha^2 x \geq 1 + x.$$

$$2\alpha + \alpha^2(1 - x) \geq 0.$$

Since $\alpha \geq 0$ (as $w_1 = a_1^2$) and $x \in [0, 1]$, the inequality holds true. The boundary cases $r = 0, r = \pm 1, \alpha = 1$ hold trivially. $\qquad \square$

**Proof that rank increases under non-linearity**   We now connect the gap bound (Lemma A.1) to the increase in Rényi-2 rank.

*Proof of Theorem 3.1 (Rank Gain and Decorrelation Potential).*  Let $C$ be a $d \times d$ correlation matrix. The Rényi-2 effective rank is $\mathrm{er}_2(C) = (\mathrm{tr}\, C)^2/\|C\|_F^2$. Since $C$ and $K_\phi(C)$ are correlation matrices, $\mathrm{tr}(C) = \mathrm{tr}(K_\phi(C)) = d$.

1. Rank Ratio: The ratio of the effective ranks is:

$$R(C) = \frac{\mathrm{er}_2(K_\phi(C))}{\mathrm{er}_2(C)} = \frac{d^2/\|K_\phi(C)\|_F^2}{d^2/\|C\|_F^2} = \frac{\|C\|_F^2}{\|K_\phi(C)\|_F^2}.$$

2. Total Decorrelation Gap $\Delta(C)$: We define the total reduction in the squared Frobenius norm:

$$\Delta(C) = \|C\|_F^2 - \|K_\phi(C)\|_F^2.$$

Since $\|M\|_F^2 = d + \sum_{i \neq j} M_{ij}^2$ for a correlation matrix $M$:

$$\Delta(C) = \sum_{i \neq j}(C_{ij}^2 - K_\phi(C_{ij})^2).$$

3. Applying the Bound: We apply the Universal Quartic Gap Bound (Lemma A.1) to each term:

$$\Delta(C) \geq \sum_{i \neq j} \gamma_\phi C_{ij}^2(1 - C_{ij}^2) = \gamma_\phi \Psi(C).$$

4. Bounding the Rank Ratio: We express the rank ratio using the gap $\Delta(C)$:

$$R(C) = \frac{\|C\|_F^2}{\|C\|_F^2 - \Delta(C)}.$$

Substituting the lower bound for $\Delta(C)$:

$$R(C) \geq \frac{\|C\|_F^2}{\|C\|_F^2 - \gamma_\phi \Psi(C)} = \frac{1}{1 - \gamma_\phi \frac{\Psi(C)}{\|C\|_F^2}}.$$

We use the inequality $(1 - u)^{-1} \geq 1 + u$, valid for $u \in [0, 1)$. Let $u = \gamma_\phi \frac{\Psi(C)}{\|C\|_F^2}$. We know $u \geq 0$. We verify $u < 1$. Since $\Delta(C) \geq \gamma_\phi \Psi(C)$, we have $u \leq \frac{\Delta(C)}{\|C\|_F^2}$. As $\|K_\phi(C)\|_F^2 \geq d > 0$, we have $\Delta(C) < \|C\|_F^2$. Thus $u < 1$.

Applying the inequality:

$$R(C) \geq 1 + \gamma_\phi \frac{\Psi(C)}{\|C\|_F^2}.$$

5. Equality Conditions: The rank ratio equals 1 if and only if $\Delta(C) = 0$. This implies $\gamma_\phi \Psi(C) = 0$. This occurs if $\gamma_\phi = 0$ (the activation is linear, $\alpha = 1$) or if $\Psi(C) = 0$ (all correlations $C_{ij} \in \{0, \pm 1\}$). $\qquad \square$

**Validation of the Quartic Gap Bound.** Figure A.1 (Top Row) validates the Universal Quartic Gap Bound (Lemma A.1). For representative activations, we plot the actual decorrelation gap $\Delta(c) = c^2 - K_\phi(c)^2$ against the theoretical lower bound $\gamma_\phi c^2(1 - c^2)$ over the correlation domain $c \in [-1, 1]$. In all cases, the actual gap strictly dominates the lower bound. The gap vanishes only at the fixed points $c \in \{0, \pm 1\}$ or for linear functions ($\gamma_\phi = 0$), confirming that nonlinear activations strictly reduce correlation magnitude (and thus increase rank) for any non-trivial input correlation structure.

**Validation of Rank Gain.** Figure A.1 (Bottom Row) tests the Rank Gain Theorem (Theorem 3.1). We generate a continuum of correlation matrices $C(\lambda)$ ranging from the identity matrix to a dense random matrix. We compute the actual Rényi-2 rank expansion ratio $R(C) = \mathrm{er}_2(K_\phi(C))/\mathrm{er}_2(C)$ and compare it to the derived lower bound $1 + \gamma_\phi \Psi(C)/\|C\|_F^2$. The results show that the actual rank gain consistently exceeds the theoretical guarantee. The gain is positive whenever the decorrelation potential $\Psi(C)$ is non-zero, demonstrating that the nonlinearity actively consumes correlation structure to generate effective rank, with an efficiency bounded by $\gamma_\phi$.

## A.2 RANK-PLASTICITY TENSION: AN EMPIRICAL VALIDATION

This section shows that maximizing the de-correlation strength $\gamma_\phi$, necessitates driving the activation function into regimes where the gradient vanishes (freezing).

In particular, we consider modulating activations by scale $a$ and bias $b$: $\psi(z) = \phi(az + b)$, and analyze the de-correlation strength $\gamma(a, b) = \frac{(\mathbb{E}[\psi'(Z)])^2}{\mathrm{Var}(\psi(Z))}$ as a function of shape parameters $\{a, b\}$.

Note that $\gamma(a, b) = (\mathbb{E}[\psi'(Z)])^2 / \mathrm{Var}(\psi(Z))$ coincides with the derivative of the corresponding correlation kernel at 0, i.e. $\gamma(a, b) = K_\psi'(0)$ after normalization. Thus, $\gamma(a, b)$ and the $\gamma_\phi$ used in Theorem 3.1 are related by a simple monotone reparameterization.

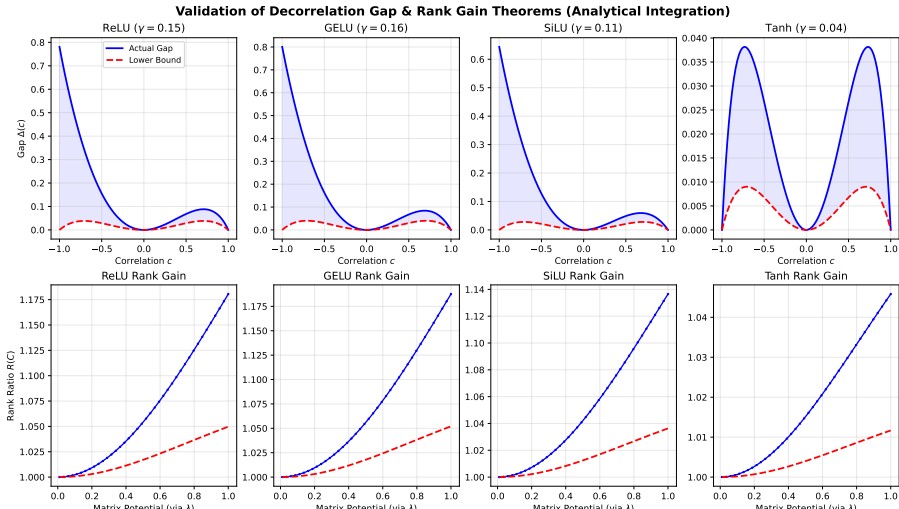

Figure A.1: **Decorrelation and rank-gain under non-linearity.** Empirical analysis activations theoretical predictions about quadratic lower bound for decorrelatioin, and for rank gain potential.

Here we empirically validate that the bias $b$ for ReLU and scale $a$ for Tanh under standard Gaussian input to measure the trade-off between decorrelation and gradient availability. As shown in Figure A.2, the decorrelation coefficient $\gamma_\phi$ is strictly monotonic with respect to the modulation parameters for both activations. Crucially, the analysis reveals a severe rank-plasticity tension: approaching maximum decorrelation ($\gamma_\phi \to 1$) necessitates driving the activation into a "frozen" regime (dead zone for ReLU or saturation for Tanh) with probability approaching $1.0$, thereby effectively eliminating the gradient flow required for training.

To extend this beyond ReLU and Tanh, Figure A.3 visualizes the fundamental trade-off described in Sec. 3. For all activations we plot the decorrelation strength $\gamma_\phi$ against the probability that the unit is frozen ($P(|f'| < \epsilon)$ where $\epsilon = 0.01$). The results confirm that to achieve high rank expansion capabilities (large $\gamma_\phi$), in most cases the parameters must push the activation into a regime where the gradient vanishes for the vast majority of inputs, demonstrating the tension between feature learning and trainability.

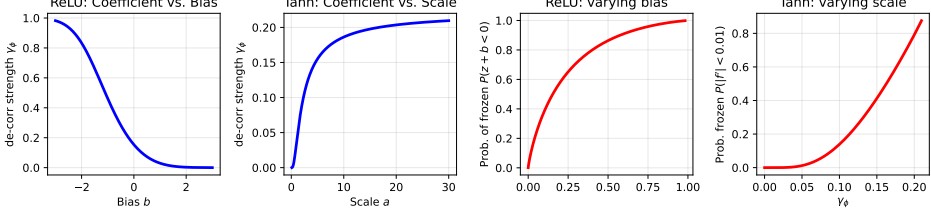

Figure A.2: **The rank-plasticity tradeoff for ReLU and Tanh.** Empirical analysis of ReLU and Tanh activations confirms that increasing the decorrelation coefficient $\gamma_\phi$ (to improve rank) strictly requires increasing the probability that the unit is frozen (dead or saturated), approaching $1.0$ as $\gamma_\phi \to 1$.

### A.3   LoP manifolds: Formal Statement and Proof

This section provides the formal definitions, statement, and proof for the LoP manifold theorem. Since the frozen manifold argument is self explanatory, we will only prove the cloning manifold result that is more non-trivial. First, let us introduce our neural network formalization. A feed-forward neural network is defined by a directed acyclic graph $G = (V, E, w)$, where $V$ is the set of nodes (neurons), $E$ is the set of directed edges (connections), $E$ representations the structure of the computational

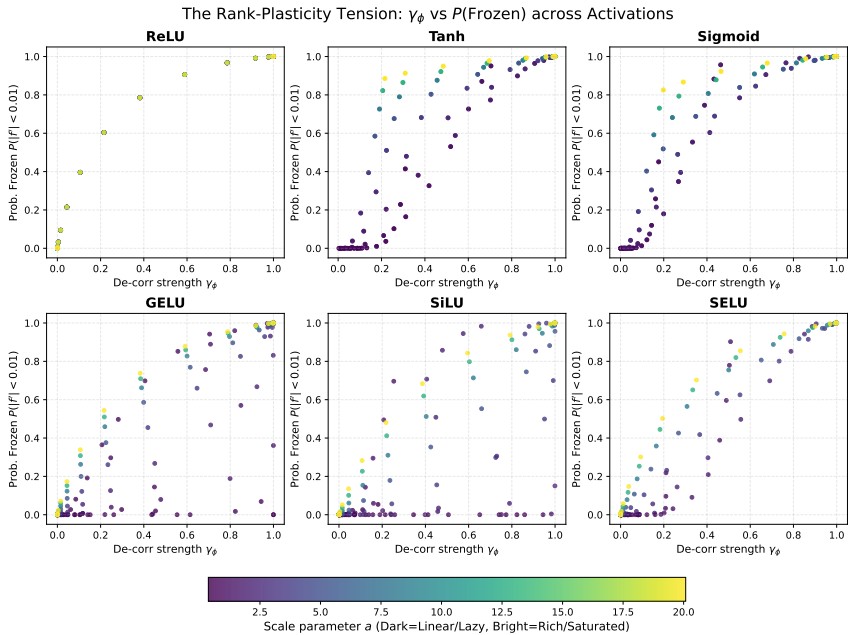

Figure A.3: **The rank-plasticity tradeoff for general activations.** Empirical analysis of all activations $\gamma_\phi$ vs the probability that the unit is frozen, when we vary the pre-activations mean and bias.

graph of the network, and $w : \mathbb{E} \to \mathbb{R}$, is the weight parameters of the network, which will be denoted $w(u, v)$ for each edge $(u, v) \in E$. Furthermore, $V_{\text{in}} \subset V$ are the input nodes, and $V_{\text{out}} \subset V$ are the output nodes. The post-activation $h(v)$ of a node $v \in V$ is computed as:

$$h(v) = \begin{cases} x_v, & \text{if } v \in V_{\text{in}}, \\ f_v\Big(\underbrace{\Sigma_{u \in \text{in}(v)} w_{u,v}\, h(u)}_{\text{pre-activation } z(v):=}\Big), & \text{otherwise.} \end{cases}$$

Here, $x_v$ is the input value for input node $v$, $f_v$ is the activation function associated with node $v$, $\text{in}(v)$ is the set of nodes with edges towards $v$. The network output is the vector of activations $h(V_{\text{out}})$. With the formal pass formally define, we can now define our backward passes. Given a loss function $\mathcal{L}(h(V_{\text{out}}), y)$ comparing the network output $h(V_{\text{out}})$ to a target $y$, the back-propagation algorithm computes gradients via error signals $\delta(v)$. The error signal is defined recursively:

$$\delta(v) = \begin{cases} \partial\mathcal{L}(h(V_{\text{out}}), y)/\partial h(V_{\text{out}}), & \text{for output nodes}, \\ \sum_{u \in \text{out}(v)} \delta(u)\, w(v, u)\, f'_u(z(u)), & v \notin V_{\text{out}}, \end{cases}$$

where $\text{out}(v)$ is the set of nodes receiving input from $v$, and $f'_u$ is the derivative of the activation function $f_u$. The gradient of the loss with respect to a weight $w(u, v)$ is then given by $\partial\mathcal{L}/\partial w(u, v) = \delta(v)\, f'_v(z(v))\, h(u)$.

**Network partition and base network definitions.** Let $G = (V, E, w)$ be the main network. A partitioning refers to a partitioning of nodes defined as:

$$\cup_{i=1}^k S_i = V, \qquad S_j \cap S_i = \emptyset \text{ for all } i \neq j.$$

Given the partitioning, we define the base network $\widetilde{G} = (\widetilde{V}, \widetilde{E}, \widetilde{w})$ where each partition is a node, the edges are union of edges between two corresponding partitions, and weights are the sum total sum of edges divided by the number of rows:

$$\widetilde{V} := \{S_i : i \in [k]\} \qquad \widetilde{E} = \{(S_i, S_j) : S_i \times S_j \cap E \neq \emptyset\} \qquad \widetilde{w}_{ij} = \frac{1}{|S_i|} \sum_{u \in S, v \in S_j} w_{uv}.$$

We can view the base graph as a "meta" graph, whose nodes are set of nodes, and its edges correspond to set of edges of the main graph. While the node and edge definitions are standard, the weight definition is slightly deviating from one might expect from standard quotient graph definitions, where the weights are total sum without averaging. The reason for this is more specific to our construction and is there to ensure similarity of the cloned and base networks forward and backward passes.

**Definitions of Weight Manifolds**  Given a network partitioning $S_1, \ldots, S_k$, and the corresponding base graph $\widetilde{G} = (\widetilde{V}, \widetilde{E}, \widetilde{w})$, here are the manifold definitions:

- The Row-wise Equitable (RE) manifold consists of all cloned weight matrices $w$ such that for every connection $(i, j) \in \widetilde{E}$ in the base network, each block $w[S_i, S_j]$ all row-sums are equal:

$$\mathcal{M}_{RE} = \left\{ w \in \mathbb{R}^{|E|} \;\middle|\; \forall (i,j) \in \widetilde{E}, \text{ and } \forall r, r' \in S_i, \text{ it holds } \sum_{u \in S_j} w_{ru} = \sum_{u \in S_j} w_{r'u} \right\}$$

  The Column-wise Equitable (CE) manifold, consists of all cloned weight matrices $w$ such that for partitioned block $w[S_i, S_j]$, all column sums are equal:

$$\mathcal{M}_{CE} = \left\{ w \in \mathbb{R}^{|E|} \;\middle|\; \forall (i,j) \in \widetilde{E}, \text{ and } \forall c, c' \in S_j, \text{ it holds } \sum_{u \in S_i} w_{uc} = \sum_{u \in S_i} w_{uc'} \right\}$$

- The Block-wise Constant (BC) manifold consists of all cloned weight matrices $w$ such that for every block $w[S_i, S_j]$, all its elements are equal:

$$\mathcal{M}_{BC} = \left\{ w \in \mathbb{R}^{|E|} \;\middle|\; \forall (i,j) \in \widetilde{E}, \text{ and } \forall u, u' \in S_i, \forall v, v' \in S_j, \text{ it holds } w_{uv} = w_{u'v'} \right\}$$

- Finally, we can define the family of all duplicate manifolds, that are affine sub-spaces of the parameters. For any matrix with row and column equitability, $w \in \mathcal{M}_{RE} \cap \mathcal{M}_{CE}$, they shift the block constant manifold $\mathcal{M}_D$. Formally:

$$\mathbb{M}_D = \{ \mathcal{M}_D(w) \mid w \in \mathcal{M}_{RE} \cap \mathcal{M}_{CE} \}, \qquad \mathcal{M}_D(w) := \{ w + T \mid T \in \mathcal{M}_{BC} \}$$

Note that all the manifolds defined above are linear or affine sub-spaces, as their constraints are all linear. There are two important facts worth mentioning that will shed more light on the upcoming theorem.

**Remark A.1.** *Note that the dimensionality of manifolds in the family $\mathbb{M}_D$ are given by the number of blocks in $W$, as opposed to number of its elements. Thus, for example if the partitioning of units forms blocks of size $n$, we would roughly expect $1/n^2$ fewer dimensions in $\mathbb{M}_D$ than in the original full parameter space.*

Furthermore, the following remark clarifies why we define these networks as cloned networks. Because when we are on these manifolds, the clone network units form perfect copies of the base network units.

**Remark A.2.** *If $W \in \mathcal{M}_{RE}$, any unit in a block $v \in S_k$, the forward activations will be identical to the corresponding base unit $h(v) = h(\tilde{v})$, where $\tilde{v}$ is the corresponding unit in the base network to block $S_k$. If we further assume $W \in \mathcal{M}_{RE} \cap \mathcal{M}_{CE}$, we will have a similar property for the backwards $\delta(v) = \delta(\tilde{v})$.*

Let us re-state the theorem on cloning to make this section more self-contained.

**Theorem A.1** (Cloned-Unit Manifold (Re-stated))**.** *Let $G = (V, E, W)$, denote a network that is partitioned with $S_1, \ldots, S_k$. Furthermore, units within a single block share the same activation function $\forall u, v \in S_k : f_v = f_u$, and there are no edges within a block. For any input and label $(x, y)$:*

  1. *If $W \in \mathcal{M}_{CE}$, then all units in the same cluster $u, v \in S_k$ have identical forward activations $h(u) = h(v)$.*

2. *If $W \in \mathcal{M}_{RE} \cap \mathcal{M}_{CE}$, then all units in the same cluster $u, v \in S_k$ have identical backward activations $\delta(u) = \delta(v)$. Furthermore, the gradients $\partial\mathcal{L}/\partial W$ will have a block-wise constant structure, such that gradients between any two units in two blocks will be equal, i.e., for any $u, u' \in S_i$ and $v, v' \in S_j$, we have $\partial\mathcal{L}/\partial W_{uv} = \partial\mathcal{L}/\partial W_{u'v'}$.*

3. *If the model weights at initialization or any point in training touch, if they lie on a manifold from the family $W \in \mathcal{M}_D$ where $\mathcal{M}_D \in \mathbb{M}_D$, given any arbitrary batches of input label pairs used to obtain subsequent model parameters $W(t)$,, any subsequent training parameter trajectory constrained to the same manifold:*

$$W(0) \in \mathcal{M}_D \implies W(t) \in \mathcal{M}_D \qquad \mathcal{M}_D \in \mathbb{M}_D, t \text{ gradient steps}$$

*Proof of Theorem A.1 (Cloned-Unit Manifold).* The proof will be done as a series of inductions. First, let us assume that we have sorted the units in a topological order $v_1, \ldots, v_n$ which exists because the network is a directed acyclic graph. Let us further assume that input nodes appear first in this list, and that outputs as the last edges in the list. Finally, because we assume no edges inside each block between the units, let us assume that the units in the same block are adjacent in our topological sort. Thus, for any two distinct blocks $S_i \neq S_j$, we either have all nodes in $S_i$ before $S_j$ or vice versa, but cannot have a mix.

**Forward cloning.** Column-equitability assumption implies identical forward for units in the same block. The induction hypothesis is that for all $k$, any preceding unit $p \leq k$, that belongs same partition $u_p, u_k \in S_i$, will have identical forward $h(u_p) = h(u_k)$. Because cloning does not apply to input units, meaning that every unit is a separate block, the hypothesis trivially holds for all input units $k = 1, \ldots, d$ where $d$ is input dimension. Now, let us prove the induction step, assuming step $k$. Let $p \leq k$ correspond to a unit in the same block $u_p, u_k \in S_i$. Now, consider all the units that have incoming edges to these two units, which necessarily must appear before $p$. Let's consider all such units within the same block $S_j$. Because these units appear before $k$, the induction hypothesis tells us that they have identical forward. Thus, the total contribution from these units to pre-activations $z(u_p)$ and $z(u_k)$ will be proportional to sum of edge weights from units in $S_j$. Because of our construction of the ordering, all the units in $S_j$ that feed into $S_i$ must occur before them. Now, the column-equal assumption implies that the sum of weights from all these units to $u_p$ and $u_k$ must be equal weight sum. Thus, we have proven that pre-activation contribution from units in $S_j$ will be identical for $u_p$ and $u_k$. Because we chose $S_j$ arbitrarily and it could have been any block, we have proven that pre-activation these units must be identical $z(u_p) = z(u_k)$. Since they also have identical activation function, they will have identical outputs $h(u_p) = h(u_k)$. This completes the induction hypothesis for forward pass cloning.

**Backward cloning.** We want to prove that column and row-equitability assumption implies identical backward for units in the same block. The proof strategy will be highly similar to the forward cloning case, with the key difference that our induction will be backward in our ordering, starting from latest output units and then moving in backward in the list. The induction hypothesis for step $k$ is that, for al $q > k$, if they are in the same block $u_k, u_q \in S_i$, they will have identical backwards $\delta(u_k) = \delta(u_q)$. Because output units are not themselves cloned, the induction step holds trivially for the last output nodes. Now let us prove the induction hypothesis for $k$ assuming that it holds for all higher steps. Now, for some arbitrary block $S_j$ that units in $S_i$ feed into, consider all outgoing connections from $u_k, u_q$ to the units in this block. Because of our construction of the ordering, all the units in $S_j$ that $S_i$ feeds into must occur after $S_j$. Thus, by induction hypothesis, all these units must have identical backwards. Furthermore, from our row-equitability assumption we know that total edge weights from $u_k, u_q$ to these units must be identical. Thus, the summation formulas in the backward of $u_k$ and $u_q$ are similar. Finally, since $S_j$ was chosen arbitrarily, this summation is identical for all subsequent blocks, which implies the overal sum is also identical. To conclude the proof, note that because of row-equitability condition we already inherit the proof from the forward case, implying $f'(z(u_k)) = f'(z(u_q))$. Thus, both parts to the backward formula for $u_k, u_q$ will be identical, which proves they have identical backwards. This finishes the induction step.

**Gradient cloning.** This step is a straightforward consequence of the forward and backward cloning steps, and the formula that gradient of an edge from $u$ to $v$ is simply $h(u)\delta(v)$. Thus, the cloning structures in forward and backward, manifest themselves as a block structure in the gradients.

**Constrained training trajectory.** Here, the key induction step is over the gradient steps. For step $t$, the induction hypothesis is that $W(t) \in \mathcal{M}_{CE} \cap \mathcal{M}_{RE}$, and that $W(t) - W(0) \in \mathcal{M}_{BC}$. This trivially holds for initial step $t = 0$. Let us prove the induction step $t + 1$ assuming that it holds for $t$. Suppose gradient at this step $\Delta W(t)$ is defined over the loss arbitrary number of samples $\{(x_i, y_i)\}$. Because of the induction hypothesis $W(t) \in \mathcal{M}_{CE} \cap \mathcal{M}_{RE}$, our earlier results imply that the gradients for each sample $\partial \mathcal{L}_i / \partial W(t)$, will have a block-wise constant structure $\partial \mathcal{L}_i / \partial W(t) \in \mathcal{M}_{BC}$. Thus, the sum of these gradients will also have a block-wise constant structure $\Delta W(t) := \partial \mathcal{L} / \partial W(t) \in \mathcal{M}_{BC}$. Because block-wise matrices are also row- and column-equitable, this implies that the new weights will inherit those $W(t + 1) = W(t) + \Delta W(t) \in \mathcal{M}_{CE} \cap \mathcal{M}_{RE}$. Finally, our parameter shift can be written as $W(t + 1) - W(0) = \Delta W(t) + W(t) - W(0)$, where $W(t) - W(0)$ is a block-wise constant matrix and thus $W(t + 1) - W(0)$ becomes sum two block-wise constant matrices, which is itself block-wise constant $W(t + 1) - W(0) \in \mathcal{M}_{BC}$. This finishes our induction step. □

### A.4 MODULAR CLONING PROFILES AND A COMPOSITION THEOREM

This subsection formalizes a modular extension of the cloning theorem in Theorem 2.1 (see also Appx. A.3) and proves that local, module-level cloning certificates glue to yield cloning for the entire composed architecture.

**Modules, interfaces, and profiles.** A *module* is a feed-forward sub-DAG $G_M = (V_M, E_M, W_M)$ together with disjoint sets of *interface nodes* $I_M$ (inputs) and $O_M$ (outputs). We allow internal nodes $V_M^\circ := V_M \setminus (I_M \cup O_M)$ and edges that connect interface nodes to internal nodes or to other interface nodes as permitted by the DAG. Let $\widetilde{G}_M = (\widetilde{V}_M, \widetilde{E}_M, \widetilde{W}_M)$ denote a smaller *base* module.

A *cloning profile* for $M$ relative to $\widetilde{M}$ consists of surjections

$$\pi_M^{\text{in}} : I_M \twoheadrightarrow \widetilde{I}_M, \qquad \pi_M^{\text{out}} : O_M \twoheadrightarrow \widetilde{O}_M,$$

inducing partitions $\mathcal{P}_M^{\text{in}} = \{ (\pi_M^{\text{in}})^{-1}(i) : i \in \widetilde{I}_M \}$ and $\mathcal{P}_M^{\text{out}} = \{ (\pi_M^{\text{out}})^{-1}(o) : o \in \widetilde{O}_M \}$. Intuitively, all interface units in the same block are *clones* of the corresponding base port.

We say two wired modules $A \to B$ have *matching profiles* if their shared interface partitions coincide after applying the wiring map $\omega_{A \to B} : O_A \to I_B$, i.e.

$$\omega_{A \to B}(\mathcal{P}_A^{\text{out}}) = \mathcal{P}_B^{\text{in}} \quad \text{as partitions of } I_B,$$

and dually for the reversed wiring used by backpropagation. More generally, a whole network has matching profiles if this holds on every inter-module edge set.

**Module-level cloning manifold.** Fix a module $M$ with profile $(\mathcal{P}_M^{\text{in}}, \mathcal{P}_M^{\text{out}})$. Extend these interface partitions to a partition of *all* nodes $V_M$ by assigning each internal node of $M$ to the block of its corresponding base-node in the collapsed base graph $\widetilde{G}$ defined in Appx. A.3. On $M$, define the (affine) *module cloning manifold*

$$\mathcal{M}_D(M) = \{ W_M : W_M \in \mathcal{M}_{RE} \cap \mathcal{M}_{CE} \text{ with respect to the induced partition of } V_M \},$$

i.e., each inter-block weight submatrix is row- and column-equitable (block-wise constant up to redistribution), reusing the notation of Appx. A.3. This generalizes the block-constant manifold $\mathcal{M}_{BC}$ by allowing intra-block redistribution while preserving equal in/out block-sums.

**Definition A.1** (Module-level cloning certificate). *A module $M$ endowed with profile $(\mathcal{P}_M^{\text{in}}, \mathcal{P}_M^{\text{out}})$ admits a* cloning certificate *if the following hold for every batch:*

(MC1) **Forward interface preservation.** *If inputs in the same block of $\mathcal{P}_M^{\text{in}}$ carry identical values, then for any $W_M \in \mathcal{M}_D(M)$ all outputs in the same block of $\mathcal{P}_M^{\text{out}}$ are identical (forward cloning).*

(MC2) **Backward interface preservation.** *If the output adjoints (backprop signals) are blockwise identical on $\mathcal{P}_M^{\text{out}}$, then for any $W_M \in \mathcal{M}_D(M)$ the input adjoints are blockwise identical on $\mathcal{P}_M^{\text{in}}$ (backward cloning).*

(MC3) **Gradient closedness.** *Under (MC1)–(MC2), the per-edge gradient $\partial \mathcal{L} / \partial W_M$ is block-wise constant on each inter-block submatrix, hence $\nabla \mathcal{L}(W_M)$ is tangent to $\mathcal{M}_D(M)$ and first-order parameter updates initialized on $\mathcal{M}_D(M)$ remain on $\mathcal{M}_D(M)$.*

**Remark A.3** (Optimizers covered). *(MC3) implies closure under any first-order optimizer whose update is a (possibly stateful) scalar multiple of the local gradient on each parameter and whose internal state is identical across clones at initialization (e.g., SGD, momentum, RMSProp, Adam with tied clone states). Weight decay that acts per-parameter independently may break exact symmetry; see Appx. A.3. Dropout violates (MC1)–(MC2) because independent masks destroy blockwise equality in the forward/backward signals.*

**Lemma A.2** (Module certificate from $\mathcal{M}_{RE} \cap \mathcal{M}_{CE}$). *If $W_M \in \mathcal{M}_{RE} \cap \mathcal{M}_{CE}$ for the induced partition of $V_M$, then $M$ satisfies (MC1)–(MC3).*

*Proof.* This is the restriction of Theorem 2.1 to the subgraph $G_M$ with its node partition: row-equitability yields identical forward values within blocks, column-equitability yields identical backward adjoints within blocks, and $d\mathcal{L}/dW_M = h\,\delta^\top$ is block-wise constant across inter-block submatrices. Tangency of the gradient to $\mathcal{M}_{RE} \cap \mathcal{M}_{CE}$ follows exactly as in Appx. A.3. $\qquad\square$

**Theorem A.2** (Composition theorem for modular cloning). *Let a feed-forward network be formed by wiring modules $\{M_\ell\}_{\ell=1}^{L}$ with matching profiles at every interface. Suppose each $M_\ell$ admits a cloning certificate (Def. A.1) and that parameters are initialized on the product manifold $\prod_\ell \mathcal{M}_D(M_\ell)$. Then:*

1. ***Global forward cloning.*** *If the external inputs respect the input profile of the first modules, then all internal interfaces and the final outputs are blockwise identical according to the propagated profiles. Equivalently, the composed network is a cloned enlargement of the composed base network.*

2. ***Global backward cloning.*** *For any loss, if the final output adjoints are blockwise identical, then all internal interface adjoints and the external input adjoints are blockwise identical according to the propagated profiles.*

3. ***Persistence under training.*** *The network gradient is tangent to $\prod_\ell \mathcal{M}_D(M_\ell)$, hence any first-order parameter update that preserves (MC3) at the module level preserves the global cloning manifold and items 1–2 continue to hold at all subsequent steps.*

*Proof. Forward.* Order modules topologically. Assume the external inputs are blockwise identical on the first-layer profiles. Applying (MC1) to the first module yields blockwise-identical outputs on its output profile. By profile matching, these outputs equal the input profile of the next module, so (MC1) applies again. Induction over modules yields blockwise equality at every interface and at the final outputs.

*Backward.* Reverse the topological order. Start from blockwise-identical adjoints at the final outputs. By (MC2) for the last module, the incoming adjoints to its inputs are blockwise identical. Profile matching identifies these with the previous module's output profile, so (MC2) applies again. The inductive step propagates back to the external inputs.

*Persistence.* By (MC3), in each module the gradient is block-wise constant on inter-block submatrices, i.e., tangent to $\mathcal{M}_D(M_\ell)$. The product of affine manifolds is an affine manifold with tangent equal to the product of tangents, so the global gradient is tangent to $\prod_\ell \mathcal{M}_D(M_\ell)$. Thus first-order updates initialized on this product manifold remain on it, and the previous two items re-apply at every step. $\qquad\square$

**Remark A.4** (Coverage: modern architectures). *The certificate (Lemma A.2) is satisfied by the standard width/channel/heads expansions used in practice:*

- ***MLPs / Linear layers:*** *Duplicate hidden units; enforce RE/CE by tiling weights with appropriate $1/(input\ expansion)$ scaling; duplicate biases. Matches the implementation in* `clone_linear`.

- ***CNNs / Conv layers:*** *Duplicate channels (in/out); tile kernels with $1/(input\ expansion)$ scaling; duplicate biases (`clone_conv1d`, `clone_conv2d`). Spatial pooling is per-channel and thus profile-preserving.*

- ***Normalization:*** *BN/LN/GN with duplicated $(\gamma, \beta)$ and running stats per clone are profile-preserving (`clone_normalization`).*

- ***Activations and elementwise ops:*** *Elementwise maps are profile-preserving (`clone_activation`); parameter-free ops are trivially preserved (`clone_parameter_free`).*

- ***ResNets:*** *Residual addition preserves cloning provided both branches use the same profile; block-level expansions meet RE/CE at each addition.*

- ***Transformers/ViTs:*** *(i) Embedding/patch-projection expansions via tiling (`clone_embedding`); (ii) Multi-head attention via head duplication; per-head linear maps satisfy RE/CE; concatenation is a profile-preserving reshape; (iii) MLP sub-blocks as in MLPs; (iv) LayerNorm is profile-preserving. The `CloneAwareFlatten` operator ensures profile-preserving reshapes between conv/linear stages.*

*By contrast, **Dropout** with independent masks across clones breaks (MC1)–(MC2) and thus is excluded from this corollary (see also discussion in the main text).*

**Remark A.5** (Minimal check-list for a new module). *To certify a new module $M$:*

1. *Choose interface partitions $(\mathcal{P}_M^{\mathrm{in}}, \mathcal{P}_M^{\mathrm{out}})$ and extend them to $V_M$.*

2. *Verify $W_M \in \mathcal{M}_{RE} \cap \mathcal{M}_{CE}$ for the induced partition (row/column equitability per inter-block submatrix).*

3. *Conclude (MC1)–(MC3) by Lemma A.2.*

4. *Ensure adjacent modules use matching profiles at shared interfaces.*

*Under these conditions, Theorem A.2 guarantees network-level cloning and its persistence under training.*

**Observation A.1** (Connection to the implementation). *The functions `clone_{linear,conv1d,conv2d,normalization,embedding,activation}` and `model_clone` implement the RE/CE tiling and profile-preserving reshapes described above, while `test_activation_cloning` empirically verifies (MC1)–(MC2) layer-wise via forward/backward $R^2$. The `CloneAwareFlatten` operator is a profile-preserving connector that keeps duplicated channels adjacent, ensuring that profiles match across CNN→FC boundaries.*

## A.5 STABILITY OF LoP MANIFOLDS.

While Theorem 2.1 establishes the *existence* of LoP manifolds under exact conditions (perfect saturation, perfect cloning), in practice, these conditions might only be approximately reached during training. This leads to the question of whether near-LoP states will move back closer to the LoP manifold under gradient descent dynamics, or will they move away from it. To address this, we introduce the notion of the stability of an LoP manifold.

**Definition A.2** (Stability of LoP Manifold). *Let $\mathcal{M}$ be an LoP manifold and $N_\theta \mathcal{M}$ be the normal space to $\mathcal{M}$ at $\theta \in \mathcal{M}$. The stability of $\mathcal{M}$ is characterized by the Hessian $\nabla_\theta^2 \mathcal{L}(\theta)$ in directions normal to $\mathcal{M}$:*

- ***Stable LoP:*** $\forall v \in N_\theta \mathcal{M} \setminus \{0\} : v^\top \nabla_\theta^2 \mathcal{L}(\theta) v > 0.$ *(Perturbations revert to LoP)*

- ***Unstable LoP:*** $\forall v \in N_\theta \mathcal{M} \setminus \{0\} : v^\top \nabla_\theta^2 \mathcal{L}(\theta) v < 0.$ *(Perturbations escape LoP)*

- ***Saddle LoP:*** $\exists v_1, v_2 \in N_\theta \mathcal{M}$ *s.t.* $v_1^\top \nabla_\theta^2 \mathcal{L} v_1 > 0$ *and* $v_2^\top \nabla_\theta^2 \mathcal{L} v_2 < 0.$ *(Escape is direction-dependent)*

**Remark A.6.** *Stability in the normal space to the manifold (convexity of the loss in these directions) does not imply that the loss is convex in general (i.e., also within the manifold or in other directions). These conditions are local characterizations of the loss landscape geometry around the manifold.*

To understand the practical implications of these stability types, consider injecting a small perturbation $\Delta\theta$ that pushes the parameters $\theta$ slightly off the manifold $\mathcal{M}$. If $\mathcal{M}$ is stable, the subsequent gradient steps $-\nabla \mathcal{L}(\theta + \Delta\theta)$ will tend to project back towards $\mathcal{M}$. If $\mathcal{M}$ is unstable, these steps will tend to move further away. For a saddle LoP manifold, escape depends on the direction of the initial

perturbation relative to the eigenvectors of the Hessian in the normal space. Therefore, the strongest form of LoP corresponds to a *stable* LoP manifold, as it actively resists escape. An unstable manifold is the easiest to escape. A saddle manifold presents a mixed scenario, where random perturbations may or may not escape depending on the perturbation vector being in a positively or negative space orientation.

# B  EMPIRICAL APPENDIX

This section provides comprehensive details of the experimental setups, additional empirical results, figures supporting claims made in the main text, and visualizations.

## B.1  EXPERIMENTAL DETAILS

This section outlines the experimental setup, methodologies, and general procedures employed for the empirical analysis of Loss of Plasticity (LoP) in neural networks.

### B.1.1  OVERVIEW OF EXPERIMENTAL PARADIGMS

Our investigation into LoP encompasses three primary experimental paradigms.

**Continual Learning Experiments**  These experiments involve training models on a sequence of temporally independent tasks where data from previously learned tasks is unavailable. Tasks are typically formulated by partitioning the output classes of standard datasets, and for any given task $t$, the model is trained exclusively on its assigned class subset $\mathcal{C}_t$. We trained our models on Tiny ImageNet, which consists of 200 classes, by creating a sequence of 40 tasks, each containing a disjoint subset of 5 classes. Each task is trained for 500 steps, and validation is performed periodically, resulting in 20,000 total training steps. The training protocol included optional reinitialization of model output layer weights and biases are reset to zero before starting each new task to mitigate interference.

**Neural Network Cloning Experiments**  These experiments study the effects of neuron duplication using a two-stage training protocol. Initially, a base model is trained on a target task to establish baseline performance. Subsequently, this base model is expanded by a specified factor (always fixed to two), using the cloning procedures detailed later. The expanded (cloned) model is then trained. To compare the base and cloned model, we also keep training the base model at the same time during this second phase. The results presented here are all on the CIFAR-10 dataset, and we used 20 epochs to train the base model and 500 epochs to train the cloned model. Functional equivalence post-cloning is verified by ensuring the cloned model produces activations identical to its base, assessed via $R^2$ scores between corresponding layer activations. $R^2$ scores, computed for each layer, measure if the mean of cloned units can explain the variance of all units in that block.

**Bit Flipping Experiments**  These experiments simulate a slowly-changing regression problem to evaluate network adaptability to gradually drifting input distributions. An illustrative benchmark for studying adaptability is the 'bit-flipping' experiment, an online regression task where the model receives an $m$-bit input vector $x$ and must predict an output $y$. The environment is non-stationary: a subset of $f$ input bits are designated 'flipping bits,' and at regular $T$-step intervals, one of these $f$ bits is randomly inverted. The remaining $m - f$ input bits are randomly sampled at each step. The target output $y$ is generated by a fixed (but unknown to the learning model) two-layer network, and a two-layer MLP is trained to learn this continuously drifting target function. The complexity of the learning model is typically designed to be less than that of the data-generating process, thereby creating a challenging scenario for maintaining plasticity. A target network with Linear Threshold Units (LTUs) implements $h_i = \text{LTU}(w_i^T x - \theta_i)$ and $y = w_{\text{out}}^T h + b_{\text{out}}$. A Linear Threshold Unit operation is defined as $\text{LTU}(z) = 1$ if $z \geq 0$, and 0 otherwise (a Heaviside step function). For the target network, the specific thresholds are $\theta_i = (m \cdot \beta) - S_i$, and $S_i = \sum_{j:w_{ij}<0} 1 - 0.5 \cdot w_{i,m+1}$. Input consists of $m$ bits plus a bias bit; $f$ of these bits are "flipping bits" changing every $T$ time steps (one randomly selected flipping bit is inverted), while the remaining $m - f$ bits are randomly sampled each step. A two-layer MLP with a configurable activation function is trained online to learn this target.

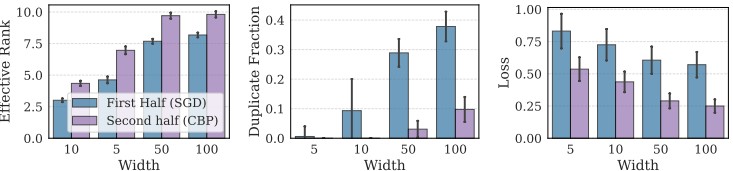

Figure B.1: Bit Flipping experiment on 5M samples, switching from SGD to CBP at 2.5M samples. Low rank structures emerge during training with standard Backpropagation (SGD), but after the switch Continual Backpropagation (CBP) is able to recover representational diversity, suggesting that CBP-like training could be effective for cloning too. (Experimental details in Appx. B).

Figure B.2: Bit Flipping experiment on 5M samples, switching from SGD to CBP at 2.5M samples. For each of the two phases, we show the average over the last 100K steps. Duplicated structures (indicated by fraction of duplicate features at different layers/scales) emerge during training with standard Backpropagation (BP) but Continual Backpropagation (CBP) is able to decouple the cloned units. As the model width is increased more duplicate features emerge. The size of the data generating function is 100. (Other experimental details in Appx. B).

### B.1.2 CORE METHODOLOGIES AND IMPLEMENTATIONS

Several core methodologies underpin our experiments.

**Cloning Implementation.** Our cloning implementation is modular. For each architecture, we first need to decide the "free" parameter to expand. This is the feature dimension for MLP and ViT, and channels for CNN and ResNet. After creating a base and expanded model, our cloning implementation proceeds in a modular fashion. The key implementation idea that allowed this modular design is the principle that the cloning profile of inputs and outputs of different modules must be consistent. For example if inputs A and B to a module are assumed to be cloned, and if these are output by a different modules, that module must ensure this cloning. We can think of this as a matching cloning profile between connected modules. With this design in mind, for linear layers, weights and biases are replicated according to input/output expansion factors; weights connected to cloned input neurons are scaled (e.g., by $1/\alpha_{\text{in}}$ for an input duplication factor of $\alpha_{\text{in}}$) to maintain activation magnitudes. Convolutional layers see similar expansion of input/output channels, with kernels tiled and appropriately scaled while preserving spatial dimensions. For normalization layer, if affine features are learned, their cloning will be a simple duplication for different cloned units. The same applies to modules such as patch embeddings, which require a simple duplication. Parameterized activations (e.g., PReLU) have their parameters correspondingly duplicated or broadcast. Any other units that does not have parameters, such as softmax layer or activations without parameter, will not require any particular treatment, because it has the potential to create cloning profiles that do not match. To fix this, we implemented a clone-aware flattening operation in CNNs ensures duplicated channels remain adjacent after flattening to preserve structure for subsequent fully-connected layers.

**Noisy SGD optimizer** introduces Gaussian noise $\epsilon_t \sim \mathcal{N}(0, \sigma_t^2 \|g_t\|^2 I)$ to gradients $g_t$, where the noise scale $\sigma_t = \sigma_0 \cdot \lambda^t$ decays over time $t$ from an initial value $\sigma_0$. The values of $\sigma_0$ and $\lambda$ are hyperparameters of the optimizer. Later, we show the effect of varying them on the cloned model dynamics.

**Continual Backpropagation (CBP)** , implemented in `src/utils/cbp_optimizer.py` following the Generate-and-Test framework, aims to maintain plasticity by selectively replacing low-utility neurons. Utility tracking involves measures like *Contribution Utility* $((u_{\text{contrib}}^{(t)})_i =$

$|h_i^{(t)}| \cdot |\bar{w}_{\text{out},i}|$) and *Adaptable Contribution* $((u_{\text{adapt}}^{(t)})_i = \frac{|h_i^{(t)} - \bar{h}_i^{(t)}| \cdot |\bar{w}_{\text{out},i}|}{|\bar{w}_{\text{in},i}|})$, where $h_i^{(t)}$ is activation, $\bar{h}_i^{(t)}$ is its running average, and $\bar{w}$ terms are mean weight magnitudes. Instantaneous utilities are smoothed using an exponential moving average ($\rho$ is decay rate, $a_i^{(t)}$ is neuron age): $u_i^{(t)} = \rho u_i^{(t-1)} + (1 - \rho)\tilde{u}_i^{(t)}$, with a bias-corrected version $\hat{u}_i^{(t)} = u_i^{(t)}/(1 - \rho^{a_i^{(t)}})$. Neuron replacement occurs for eligible mature neurons ($a_i > \tau_{\text{maturity}}$) with the lowest utility (a fraction $r_{\text{replace}}$ of layer neurons $N_L$). Selected neurons are reinitialized (incoming weights via Kaiming, outgoing to zero, utility/age reset). A bias correction ($b_{\text{next}} \leftarrow b_{\text{next}} + W_{\text{out}}[:, i] \cdot \bar{h}_i$) is applied to the subsequent layer.

**Metrics for Analysis**  Our comprehensive metric suite quantifies various aspects of network behavior and plasticity loss. *Single and pair feature metrics* include the fraction of "dead" neurons, identified when $\frac{1}{N}\sum_{i=1}^{N} \mathbf{1}[|H_{ij}| < 10^{-7}] > \tau_{\text{dead}}$ for neuron $j$ across $N$ samples, with $\tau_{\text{dead}} = 0.95$. "Duplicate" neurons are detected through cosine similarity patterns, with neurons $j, k$ are considered duplicates if $\tilde{H}_j^T \tilde{H}_k > \tau_{\text{corr}} = 0.95$, where activations are normalized by feature $\tilde{H}_j = H_{\cdot,j}/\|H_{\cdot,j}\|_2$. "Saturated" neurons are identified when the ratio of gradient magnitude to mean activation magnitude $|G_{ij}|/\max(\mu_j, \epsilon)$ falls below $\tau_{\text{sat}} = 10^{-4}$ for more than $p_{\text{sat}} = 99\%$ of samples in a batch. *Representation diversity metrics* include effective rank, computed as $\exp(-\sum_i p_i \log p_i)$ where $p_i = \sigma_i/\sum_j \sigma_j$ are normalized singular values from the activation matrix SVD; stable rank, calculated as $\|\tilde{H}\|_F^4/\text{tr}((\tilde{H}^T\tilde{H})^2)$ for mean-centered activations $\tilde{H}$; *Cloning quality* is assessed by $R^2$ scores between base and cloned model activations, computed as $R^2 = 1 - \text{Var(residuals)}/\text{Var(total)}$ where the predictor is the mean of $N$ cloned units and we measure explained variance across individual units relative to the total variance in that layer. This is done for both forward and backward activations across all layers, and numbers presented here are averages across all layers and both forward and backwards for the fixed batch that we are measuring the metrics. We also keep tracking all metrics for both base and cloned model after training to provide a comparison between the two.

### B.1.3 GENERAL SETUP AND PROCEDURES

**Model Architectures**  include Multi-Layer Perceptrons (MLPs), Convolutional Neural Networks (CNNs), ResNets, and Vision Transformers (ViTs), with configurations (depth, width, activations, normalization layer, dropout). The default configurations are as follows: Our Multi-Layer Perceptron (MLP) consists of 5 hidden layers with 128 units each, employing ReLU activations, batch normalization applied before activation, and 20% dropout. The Convolutional Neural Network (CNN) architecture comprises 3 convolutional layers with $[64, 128, 256]$ channels respectively, using $3 \times 3$ kernels with stride 1 and padding 1, followed by $2 \times 2$ max pooling operations. The convolutional features are processed by a single fully connected layer with 512 units, with ReLU activations, batch normalization, and 10% dropout throughout. For ResNet, we implement a ResNet-18 variant with $[2, 2, 2, 2]$ residual blocks per stage, starting with 64 base channels that double at each stage, using ReLU activations, batch normalization, and 10% dropout. The Vision Transformer (ViT) architecture divides input images into $8 \times 8$ pixel patches, which are projected to 384-dimensional embeddings and processed through 6 transformer layers with 6 attention heads each. The ViT employs an MLP ratio of 4.0 (yielding hidden dimensions of 1536), GELU activations, layer normalization, and 10% dropout for both general operations and attention mechanisms. All normalization layers include learnable affine parameters ($\gamma, \beta$), unless stated otherwise, and bias terms are enabled where applicable. Default hyperparameter configurations for each architecture can be adjusted per experiment as described in the experimental setup.

**Datasets and Preprocessing**  involve standard image classification benchmarks: MNIST ($28 \times 28$ grayscale), CIFAR-10 and CIFAR-100 ($32 \times 32$ RGB with standard augmentations like random crops and flips), and Tiny ImageNet ($64 \times 64$ RGB). Standard train/test splits are used. For all the figures and results reported here, we used tiny ImageNet dataset for continual learning experiments, while for cloning, CIFAR-10 was used.

**Training Configuration**  involves optimizers like Adam or SGD without momentum and no weight decay with otherwise parameters in torch. The learning rates for the continual experiments where set

to 0.001 using Adam for all architectures except for Vision Transformer, which was set to 0.0001. For cloning experiments with dropout, we varied the learning rate on a grid 0.01, 0.001, 0.0001.

**Experimental Control** is maintained through comprehensive random seeding, which controls the randomness across all relevant libraries (Python, NumPy, PyTorch) and CuDNN deterministic mode. We used 5 seeds for all experiments to calculate confidence intervals. Experiments utilize GPUs when available, falling back to CPUs otherwise. Metrics are typically computed at fixed epoch intervals (e.g., every 5 epochs), often on consistent fixed data batches for reproducibility. Computationally intensive metrics like SVD may use subsampling of the features or samples to make them less expensive.

**Computational Resources.** For the continual learning and cloning experiments, our experimental grid consisted of approximately 2,000 individual runs (counting each random seed separately). These experiments were executed on a cluster of NVIDIA A100 GPUs, utilizing a heterogeneous mix of 40GB and 80GB memory variants. The total computational cost for these experiments was approximately 10,000 GPU-hours. The bit flipping experiments and additional theory validation experiments were conducted on a more diverse set of hardware, utilizing lower-end computational nodes equipped with NVIDIA RTX 3090, V100, and RTX 2080 GPUs. This heterogeneous setup was sufficient for these less computationally intensive experiments, and the overall compute amounted to under 100 GPU-hours on these nodes. The theoretical validation figures and numerical simulations presented in the theory appendix (Appendix A.2) were generated on a MacBook using CPU computation only.

**Figures details.** Unless stated otherwise, all our figures report standard deviations over 5 experiment randomization, by the use of a different seed. Additionally, to reduce the number of points in the plot, in Figs. 3.1, 3.2, 4.1, B.1 and B.2 we plot the average over time windows of 1000 steps.

## B.2 ADDITIONAL FIGURES AND EMPIRICAL SUBSTANTIATION

This subsection includes placeholder figures for concepts discussed in the main text, for which specific existing figures were not available or suitable for direct inclusion in the main body.

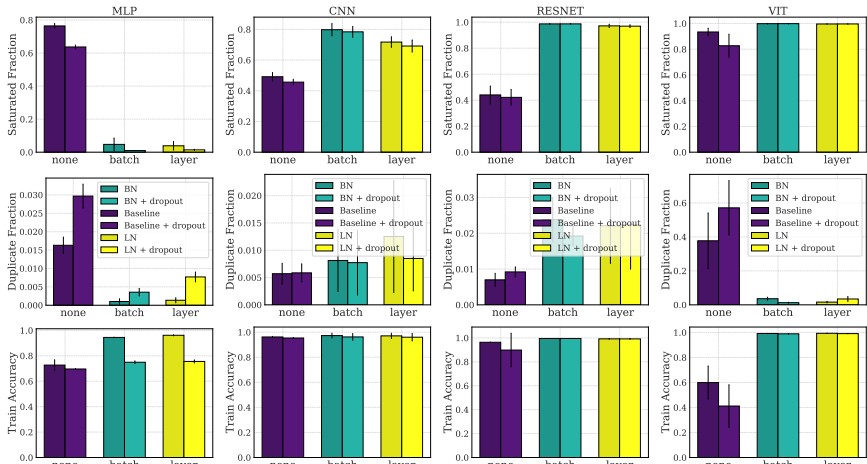

Figure B.3: Normalization reduces the number of dead/saturated units (top row) and duplicated units (middle row), and its impact on training accuracy (bottom row) across different architectures. The training accuracy displayed is calculated as the average online accuracy over the entire training length. These results highlight the role of normalization in mitigating LoP symptoms.

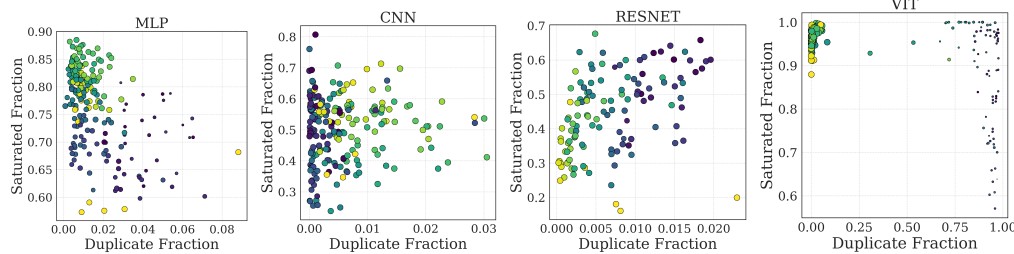

Figure B.4: Evolution of duplicate/dead unit fractions and training accuracy. The colors correspond to training steps (lighter is earlier) and the points size to the Training Accuracy (bigger is higher). This figure illustrates the correlation between the increase in LoP symptoms (duplicate/dead units) and training dynamics.

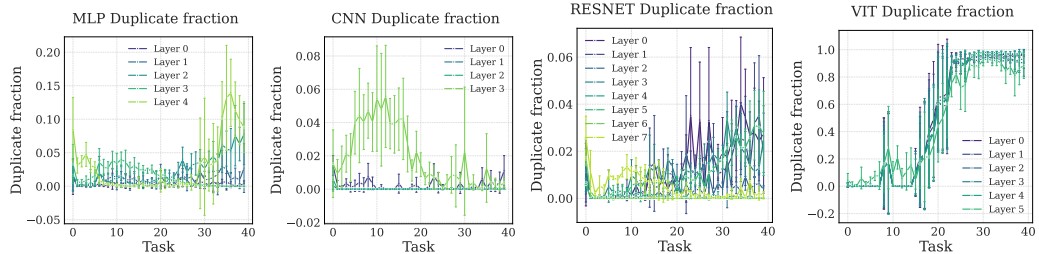

Figure B.5: Emergence of duplicate units layer-wise during training without normalization and no dropout. This figure shows the increasing fraction of duplicate units as training progresses, a symptom of LoP.

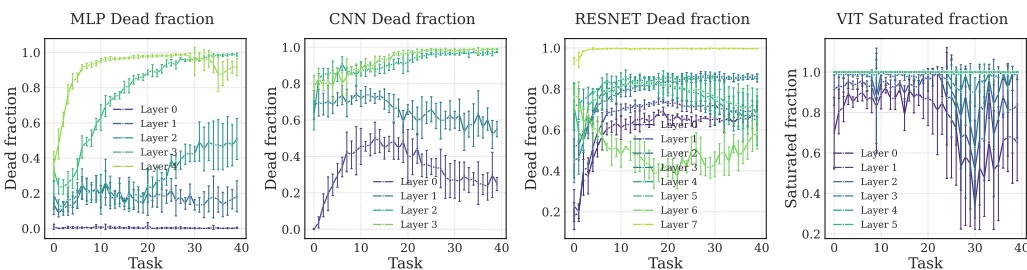

Figure B.6: Emergence of dead or saturated units layer-wise during training without normalization and no dropout. This figure shows the increasing fraction of dead units as training progresses, a symptom of LoP.

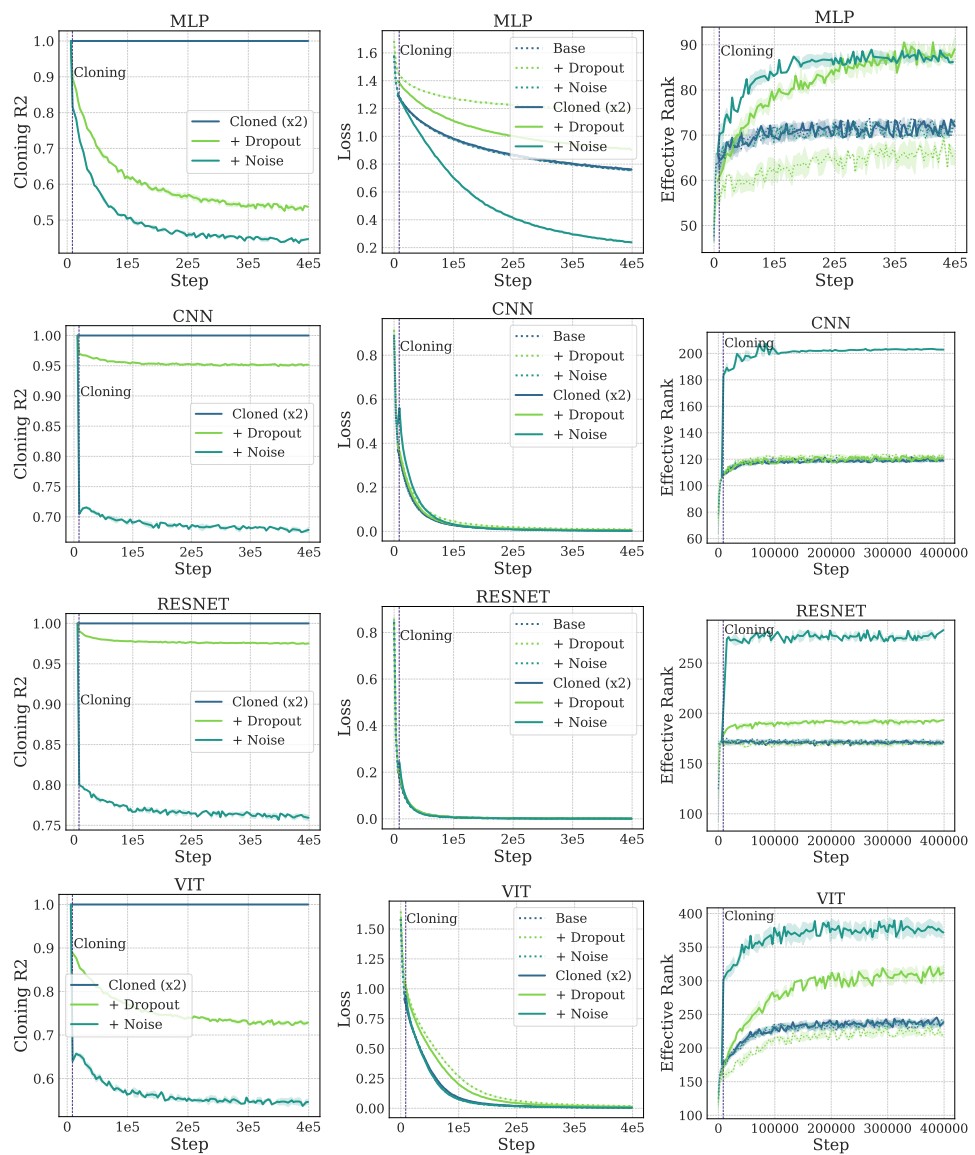

Figure B.7: Cloning experiments across architectures. Configurations details: SGD with LR=0.01, Noisy SGD with $\sigma = 0.01$ and $\lambda = 0.999$, and Dropout with probability $0.1$. Normalization used: Batch Norm for all architectures, except ViTs, where we use Layer Norm.

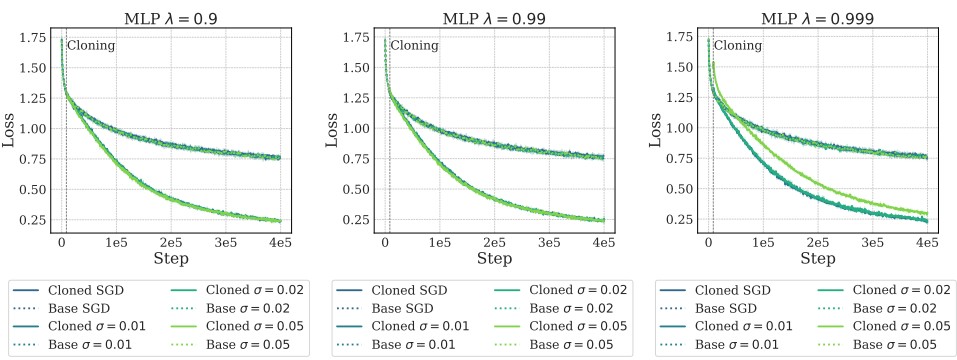

Figure B.8: Effect of noise scale parameter $\sigma$ in Noisy SGD for the Cloning MLP Experiments.

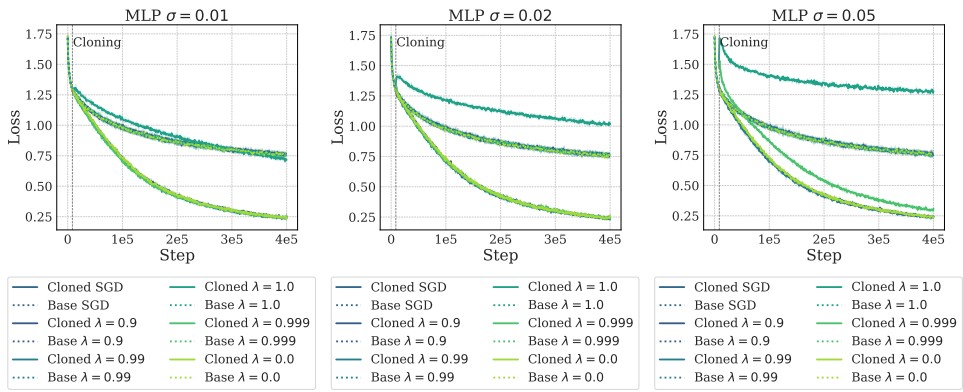

Figure B.9: Effect of noise decay parameter $\lambda$ in Noisy SGD for the Cloning MLP Experiments.

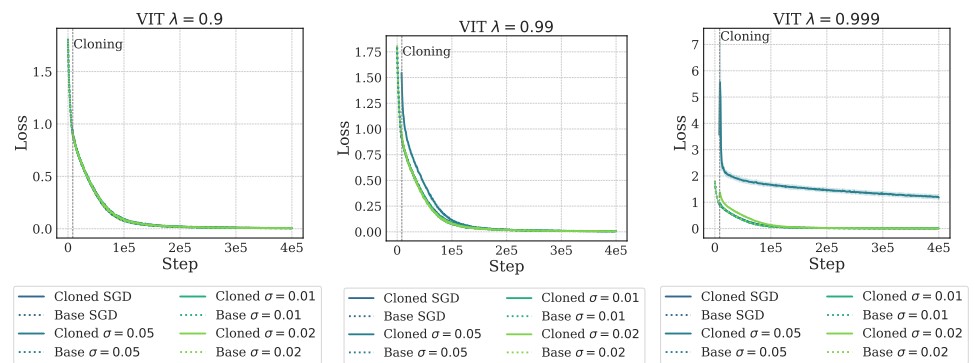

Figure B.10: Effect of noise scale parameter $\sigma$ in Noisy SGD for the Cloning ViT Experiments.

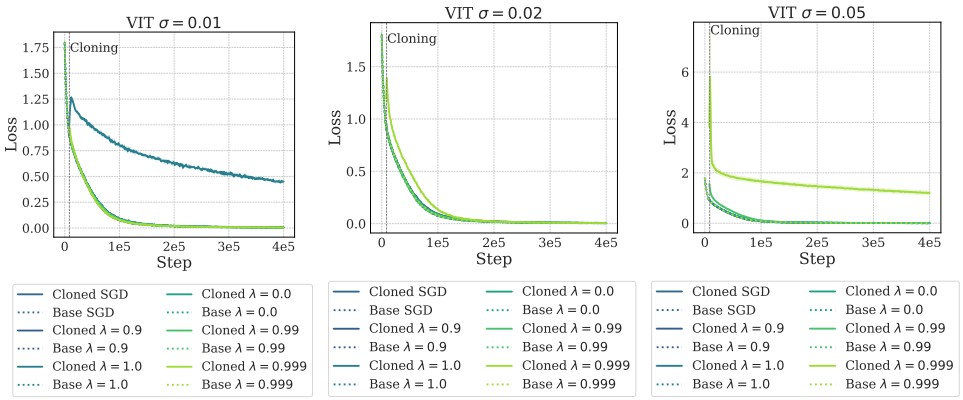

Figure B.11: Effect of noise decay parameter $\lambda$ in Noisy SGD for the Cloning ViT Experiments.

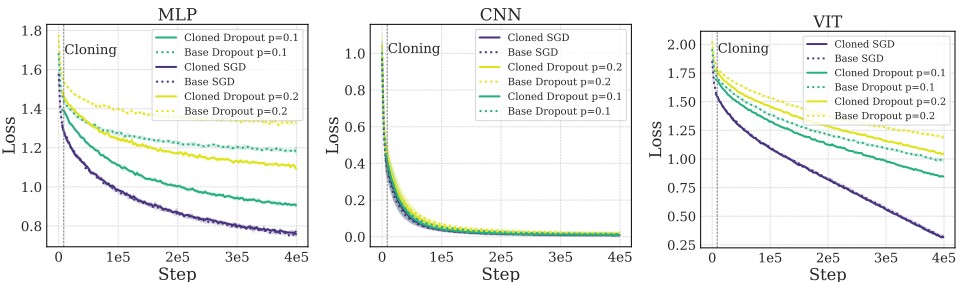

Figure B.12: Effect of dropout probability parameter for the Cloning MLP, CNN and ViT Experiments. Batch norm is used for the MLP and CNN models, and Layer norm for the ViT model.

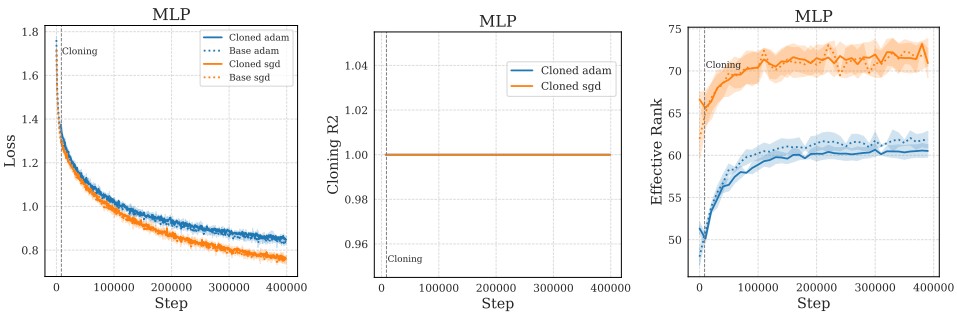

Figure B.13: Differences between SGD and Adam optimizers in the MLP Experiments. Like SGD, Adam cannot escape the base sub-manifold, although the dynamics are different.

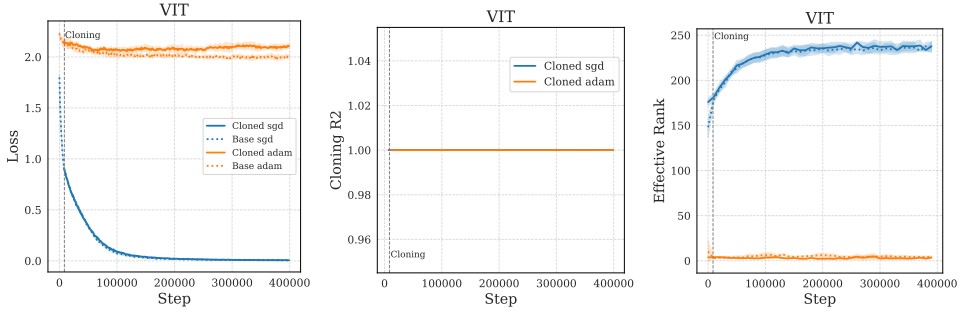

Figure B.14: Differences between SGD and Adam optimizers in the ViT Experiments. Like SGD, Adam cannot escape the base sub-manifold, although the dynamics are different.

