# OpenReview forum: "Barriers for Learning in an Evolving World:  Mathematical Understanding of Loss of Plasticity"
_ICLR.cc/2026/Conference — ICLR 2026 Poster_

### Official Review · Reviewer_kAhz · 2025-10-23

**Soundness:** 3
**Presentation:** 4
**Contribution:** 3
**Rating:** 8
**Confidence:** 3

**Summary:**

This paper provides a theoretical foundation for understanding and subsequently tackling the LoP phenomenon encountered in learning in dynamic settings. The paper pinpoints properties of solutions that inhibit the ability of the model to adapt and effectively learn in the future.

**Strengths:**

The paper is well written, has a clear structure and is easy to follow.

The paper identifies a geometrical property of the loss of plasticity manifolds that prevents gradient based methods from escaping them. Namely that the gradients are tangent to the manifold. Then they proceed to identify to categories of such manifolds that might occur while training, as the networks effective dimension decreases.

The discussion about escaping them through noise, dropout or some form of randomness is an interesting idea and apparently according to the experiments an effective one as well.

The paper also provides an explanation for the emergence of these manifolds through training, which is interesting.

**Weaknesses:**

The paper has no significant weaknesses in my opinion. As in the sense that it provides an explanation for the LoP in practical settings, from a theoretical standpoint. The explanation might be somewhat incomplete. Mostly I have some concerns which I put in the questions, as I would prefer to discuss them.

For the writing. In **line 109** there is a double the and in the appendix there is a double network in **line 646** and it should be in my opinion a represents in **line 647**.

**Questions:**

I have the following questions regarding your work:

1. Do you expect other types of manifolds to exist that cause the loss of plasticity? Apart from the frozen unit and the cloning manifold, especially given that the addition of the dropout layer is not effective consistently in non artificial experiments.
2. My second question revolves around neighboring points to the manifolds you describe. While it is true that if an algorithm reaches an exact cloning manifold then it will be impossible to escape, what happens when we are in the points around it?
3. Furthermore as a continuation to my previous question. What happens when you use SGD in these scenarios of inexact solutions. It seems hard for me to believe that if you reduce the batch size to 1 (even though this goes against general practice) there is not enough noise from the individual and new samples to help the model escape the neighborhood of the manifold. I am curious about this question since SGD with Gaussian noise manages to escape these manifolds.
4. Is the fact that we arrive at a neighborhood of the manifold and not the exact manifold a potential reason for the differentiation between the behavior of dropout layers in the artificial experiment and the non-artificial one?

---

> ### Author Response · Authors · 2025-11-28
>
> We thank the reviewer for their positive evaluation of our work, and for their highly insightful comments and questions.
>
> We provide the responses to their questions below.
>
>
> >  For the writing. In line 109 there is a double the and in the appendix there is a double network in line 646 and it should be in my opinion a represents in line 647.
>
> > Fixed
>
> > Questions:
> > I have the following questions regarding your work:
> Do you expect other types of manifolds to exist that cause the loss of plasticity? Apart from the frozen unit and the cloning manifold, especially given that the addition of the dropout layer is not effective consistently in non artificial experiments.
>
> This is a very insightful and intriguing question. We hypothesize that the most persistent LoP manifolds are indeed likely to be linear or affine (like the frozen and cloned unit manifolds we identified) due to the discretization inherent in training. While following the direciton of Gradient Flow stays invariant on any manifold where the gradient is tangent, discrete Gradient Descent (GD, SGD, Adam, etc) takes straight-line steps along the tangent space. This  leads to a key distinction between affine and curved manifolds:
>
> On a curved manifold, a discrete linear step along the tangent immediately moves the parameters off the manifold, which could act as a natural escape mechanism, making curved manifolds inherently unstable due to this so called 'discretization drift'. In contrast, on an affine manifold, the tangent space is aligned with the manifold itself, and a discrete step stays perfectly within the manifold. Thus, affine LoP manifolds are inherently more robust to the discrete nature of optimization than non-linear manifolds, which underlies our hypothesis.
>
> We have added Remark 2.1 in the revised manuscript that discusses on this nuance.
>
> > My second question revolves around neighboring points to the manifolds you describe. While it is true that if an algorithm reaches an exact cloning manifold then it will be impossible to escape, what happens when we are in the points around it?
>
> This is another highly important and interesting question. While we did not answer the question theoretically, our empirical evidence pains a complex picture about stability near manifold. In case of cloned networks, most networks seem unstable, as introducing a few noisy steps to SGD is sufficient to decouple cloned units (see Figs B.8-B.12, particularly configurations with low noise magnitude and duration). But in case of cloning as a result of prolonged training the noise/perturbation required is much stronger (similar to what continual backprop does very invasively). Finally, we also notice a clear distinction between architectures, with some showing less or higher resisistance to noise injected to SGD in their ability to escape the LoP manifold.
>
> > Furthermore as a continuation to my previous question. What happens when you use SGD in these scenarios of inexact solutions. It seems hard for me to believe that if you reduce the batch size to 1 (even though this goes against general practice) there is not enough noise from the individual and new samples to help the model escape the neighborhood of the manifold. I am curious about this question since SGD with Gaussian noise manages to escape these manifolds.
>
> We e can answer in two parts
>  - On the manifold: Theorem 2.1 holds for SGD because the gradient for any mini-batch respects the symmetry. Sampling noise alone does not break the cloning symmetry. Because gradient over a batch is the sum of individual gradients, any symmetries between units for individual sample gradients remain true for the mini-batch gradient (ie, their sum). This is discussed in Remark 2.4.
>  - Near the manifold: The role of SGD noise is complex. The authors cite work on "Stochastic Collapse" (Chen et al., 2023) (Lines 96-100), which suggests that SGD noise may actually attract dynamics towards these singular regions, rather than helping to escape them.
>
> > Is the fact that we arrive at a neighborhood of the manifold and not the exact manifold a potential reason for the differentiation between the behavior of dropout layers in the artificial experiment and the non-artificial one?
>
> Yes this intuition is exactly correct. We observe that dropout is effective in symmetry breaking in cloning experiments, yet somehow unable to break them when they occur as a result of prolonged training.
>
> We again thank the reviewer for their highly constructive feedback.

---

### Official Review · Reviewer_J4Px · 2025-10-29

**Soundness:** 3
**Presentation:** 3
**Contribution:** 3
**Rating:** 6
**Confidence:** 3

**Summary:**

This paper studies loss of plasticity (LoP), the gradual inability to learn as training progresses in continual learning settings. The authors define LoP in terms of manifolds and show theoretically that GD/SGD updates that occur in frozen-unit manifolds (from saturated units) and cloning manifolds (from duplicate/redundant units) will remain in their respective manifolds, preventing subsequent learning. They also show that if a network is composed of individual “modules” (layers or blocks) that satisfy some properties, then the entire network resides on a cloning manifold. Experimentally, they compare a base network and the same network with cloned units and show that they exhibit the same behavior as training progresses, indicating an inability to escape the cloning manifold. However, perturbations like dropout and noisy SGD can help to escape the manifold because they act as symmetry-breaking. Studying CL tasks, they show that as training progresses, the fraction of dead units and fraction of duplicated units tends to increase and effective rank and performance decrease. They also show how batch norm and layer norm can help to prevent the saturation of units (and subsequent LoP), preserving effective rank, and that training with a noisy learning rule like continual backpropagation (CBP) can help networks recover from LoP.

**Strengths:**

The paper formalizes and studies an interesting explanation for LoP. Indeed, low-rank connectivity is generally associated with better representational quality in static settings, but the paper shows how this breaks down in continual learning settings. It further highlights two ways this behavior can happen through frozen, saturated units and through cloned units. The paper is well-contextualized in the literature. The paper complements theory with several experiments in different architectures, showing the application of their results in explaining behavior in standard models and providing methods to prevent or escape LoP based on their analysis.

**Weaknesses:**

The specific contributions of the paper could be outlined more, such that specific results are highlighted. There are many simulations but it’s somewhat difficult to remember and follow what is or isn’t shown in them, especially as many figures are in the appendix and experiments are interspersed with theory. I found some parts (like Theorem 3.1) difficult to follow. Minor point, but there are many in-text typos.

In the paper, it’s mentioned that “initial growth is followed by a compression phase” and that this is “consistent with the information bottleneck principle.” However, the existence of a compression phase in neural network training is disputed and Saxe et al., 2018 showed that the central claims of Schwart-Ziv and Tishby, 2017 did not hold true in the general case. I’m thus skeptical of the claim of a compression phase in training.

Related, although the authors briefly mention the NTK and rank, the paper could link its results and study to feature learning more explicitly as it is very related to their analysis. For example, low-rank connectivity is typically related to the so-called rich learning regime, while higher-rank connectivity is related to lazy learning. Hence, the presence of frozen units or cloned manifolds may be a symptom of rich learning, and might be prevented by a network in the lazy learning regime. This could further highlight whether and how learning regime plays a role in LoP.

I acknowledge that the authors do show experiments of cases where there are duplicate or dead neurons and correlate this to worsened learning. However, I’m somewhat skeptical that GD/SGD always pushes the network to these manifolds, and whether they might be other underlying factors related to LoP? For example, in Figure B.5 and B.6, the behavior across different architectures and layers varies dramatically and the fraction of duplicate units is quite small. Nevertheless, I think the contribution of the paper still stands on its own but there is more to be learned in future work.

**Questions:**

1. In Figure 2.1, why does adding dropout worsen performance if it manages to reduce cloning and increase effective rank?

2. I understand that dropout provides symmetry breaking, but I’m wondering if you can discuss why this increases rank and decreases cloning? Dropout increases redundancy and is used as a form of regularization, so it’s surprising to me that it would have this effect. Is it because dropout is applied after cloning? Would LoP be worsened if it were applied at the onset of training?

3. In Figure B.3, batch and layer norm only appear effective in removing saturated units for the MLP and duplicated for the MLP and VIT. Can you discuss why this behavior is not consistent across all architectures?

Suggestion: you could perhaps use weight decay to prevent saturation of units/formation of dead units?

---

> ### Author Response · Authors · 2025-11-28
>
> We would like to sincerely thank the reviewer for their highly insightful remarks and feedback about our work. Here are our point-by-point responses.
>
> > The specific contributions of the paper could be outlined more, such that specific results are highlighted.  ...
>
> Thank you for raising this important point. To address your point about lack of clarity on our contributions, we have revised the last two paragraphs  of the introduction (lines 045-063 in the revised version) to sharpen the contrast with literature and clarify our contributions, and added a new bullet point of contributions.
>
> > ...  the information bottleneck principle ... is disputed
>
> We appreciate this very insightful comment. We agree that the universality of the Information Bottleneck compression phase is disputed (e.g., Saxe et al., 2018). Accordingly, we have revised the manuscript to ground our argument in the Neural Collapse phenomenon (Papyan et al., 2020) instead. We have revised the relevant part of the paper to reflect this (see beginning of Section 3, line 232-onwards in the revised version).
>
> > ... low-rank connectivity is typically related to the so-called rich learning regime, while higher-rank connectivity is related to lazy learning .
>
> This is very intriguing and insightful comment. We  have added remark 3.1 in the revised manuscript to expand on this connection.
>
> > .. I’m somewhat skeptical that GD/SGD always pushes the network to these manifolds, and whether they might be other underlying factors related to LoP?
>
> This is a very important question raised by the reviewer. Our takeaway is captured in section 3, which argues that the underlying dynamics of feature learning and compression (neural collapse) naturally lead to creation of frozen and duplicate units. Theorem 3.1 captures this in a lower bound for rank of representation under a joint linear and nonlinear layer:  de-correlation strength $\gamma_\phi$ that determines how strongly nonlinear layers increase rank, which is maximized when activations move to frozen regions (see Section A.2), and the de-correlation potential $\Psi(C)$ which becomes zero when most features (except for a few that represent classes) become aligned or anti-aligned. Taken together, this theorem suggests that feature learning dynamics create a natural force towards the frozen and duplicate manifolds.
>
> > Questions:
> > In Figure 2.1, why does adding dropout worsen performance if it manages to reduce cloning and increase effective rank?
>
> This increased difficulty in training loss with dropout reveals the trade-off between feature diversity and optimization. Dropout increases diversity (rank) but also injects noise and acts as a regularizer, which often slows convergence and increases training loss. The higher dropout rate makes convergence even slower (See Appendix B for ablations).
>
> > .. why [dropout] increases rank and decreases cloning? Dropout increases redundancy and is used as a form of regularization, so it’s surprising to me that it would have this effect.
>
> They reason for increase in rank and decrease in cloning is that applying independent dropout masks breaks the symmetry required for identical gradient updates, causing divergence between cloned units. These diverged features are not linearly dependent anymore, thus, increasing the overall feature rank, and lower cloning score (similarity) between them.
>
> > Is it because dropout is applied after cloning? Would LoP be worsened if it were applied at the onset of training?
>
> In our experiments dropout used throughout the process (from the onset, and after cloning), but For these reasons mentioned above,  if dropout was only applied after cloning it would also have worked. But if dropout was applied at the onset and not after cloning, its symmetry breaking power will be entire gone.

---

> > ### Author Response · Authors · 2025-12-03
> >
> > > In Figure B.3, batch and layer norm only appear effective in removing saturated units for the MLP and duplicated for the MLP and VIT ...
> > >   in Figure B.5 and B.6, the behavior across different architectures and layers varies dramatically
> >
> > We thank the reviewer for this nuanced observation.
> > - ViT vs MLP/ResNet/CNN in saturated/dead units: First,  to keep each architecture close to its original, we keep ViT architecture with original GeLU activation, while MLP, CNN, and ResNet use ReLU. Thus, some qualitative differences might emerge from these factors.
> > - CNN/ResNet vs MLP/ViT duplicate units: Note that to calculate duplicated units for convolutional units, we are comparing different channels, and we speculate that even if two channels are highly similar, the spatial noise could make them appear less similar, falling below the 95% correlation threshold for being defined as duplicates. Thus, as Fig B3 shows, CNN and ResNet duplicate values are always low, and the differences observed between the bars without normalization, with BN, or LN, are all within standard deviation bars.
> >
> > While the empirical picture is complex, the observations are consistent with the overall framework and do not contradict any of our theories. Thus these architectural differences are an interesting finding work follow-up studies, rather than a failure of the theory.
> >
> > > Suggestion: you could perhaps use weight decay to prevent saturation of units/formation of dead units?
> >
> > We refer to Dohare et al. (2024), who conducted extensive ablations on L2 regularization. Their results demonstrate that while L2 regularization constrains weight magnitudes , it fails to fully mitigate loss of plasticity. Specifically, networks trained with L2 regularization still exhibit a continuous increase in dead units and a decrease in effective rank over time. Sustained plasticity requires diversity injection rather than just magnitude control.
> >
> > We thank the reviewer again for their highly valuable feedback!

---

### Official Review · Reviewer_bx6B · 2025-11-01

**Soundness:** 2
**Presentation:** 2
**Contribution:** 2
**Rating:** 4
**Confidence:** 4

**Summary:**

The authors begin their work by asking what structural properties of gradient flow lead to loss of plasticity, and in response, what sorts of algorithms or architectures can be designed to mitigate this. The authors proceed by defining a notion of a loss of plasticity manifold, which is a manifold in weight space where for each point in the manifold the gradient of the loss evaluated at that point is tangent to that manifold. Or in other words, a loss of plasticity manifold is a subset of possible network from which gradient flow cannot escape. The authors prove that for manifolds characterized by dead (saturated) neurons or those characterized by duplicated features, gradient flow cannot escape, resulting in the persistence of neuron death or redundant features under gradient descent or SGD. Empirically, the authors show that noisey GD/SGD and dropout can escape these loss of plasticity manifolds and that these manifolds can arise during continual learning.

**Strengths:**

They give a mathematical formalism of loss of plasticity due to neuron death or redundant units via the lens of stable manifolds. Specifically, the authors prove that gradient flow cannot escape a loss of plasticity manifold, and this modeling of plasticity is one way of unifying two well known-correlates: neuron death and rank collapse. This also introduces a new framework for modeling plasticity loss which future researchers may build upon.

**Weaknesses:**

While the introduction of stable manifolds is novel to this field, the results are not necessarily as novel. For instance, the implications of Theorem 2.1 are often stated in most papers in this space, notably those focused on neuron resets or spectral regularization. Moreover, the implications of this theorem are easily observed via a simple analysis of the gradient of the loss when neuron death or duplicate features are present, which many existing works explicitly state. Similarly, the empirical results appear to reproduce many of the phenomena already observed. At some level, the paper reads like it is restating existing results and observations under a more abstract formalism of stable manifolds.
The paper does not analyze theoretically the stability of empirically observed LoP manifolds.
While the paper is generally well written, it could make use of better sign-posting or a clear summary of contributions in the introduction. The motivating question is quite vague and only half way through the main body does the reader begin to understand the scope of contributions.

**Questions:**

In the bit-flipping benchmark, you switch from SGD to CBP at half-point and see recovery. Can you show sensitivity to the switch point?

---

> ### Author Response · Authors · 2025-11-28
>
> We would like to thank the reviewer for their detailed review of our work and their valuable and constructive feedback.
>
>  In the revised manuscript we have made our contributions more clear in the introduction, with a bullet point list of contributions, and have added a section to address their point about lack of clarity on contributions, and added a small experiment on timing of bit flipping (Fig B1).
>
> Here we provide our point-by-point responses:
>
> > Weaknesses:
> While the introduction of stable manifolds is novel to this field, the results are not necessarily as novel. For instance, the implications of Theorem 2.1 are often stated in most papers in this space, notably those focused on neuron resets or spectral regularization. Moreover, the implications of this theorem are easily observed via a simple analysis of the gradient of the loss when neuron death or duplicate features are present, which many existing works explicitly state. Similarly, the empirical results appear to reproduce many of the phenomena already observed. At some level, the paper reads like it is restating existing results and observations under a more abstract formalism of stable manifolds.
>
> As detailed in the manuscript, our theoretical framework fills a specific gap not addressed by the cited literature. We respectfully direct the reviewer to the following sections where these distinctions are explicitly formalized. Here we iterate to respectfully remind the reviewer of these distinctions with the literature:
>
> **Contrast with Empirical Mitigation (Mechanism vs. Symptom)**
> While prior works (e.g., Nikishin et al., 2022; Sokar et al., 2023) identify symptoms (dead units) and propose fixes, they do not derive the dynamical systems cause. Our **Definition 2.1** and **Theorem 2.1** provide this missing derivation, proving that the gradient flow becomes tangent to these manifolds ($\nabla \mathcal{L} \in T_\theta \mathcal{M}$), thereby rendering standard optimization mathematically incapable of escape without external intervention.
>
> **Contrast with Singularity**: As noted in **Remark 2.2 **, prior analyses of singularities (e.g., Chen et al., 2023; Fukumizu & Amari, 2000) generally define cloned units via **strict duplication** (where weights are identical, $w_i=w_j$). In contrast, **Theorem 2.1** proves invariance under the mathematically broader condition of **Equitable Partitions** (equal row/column sums), where individual weights may differ. This defines a broader class of LoP traps ($\mathcal{M}_C$) than the "invariant sets" previously established.
>
>
> We have found no prior work that covers our contributions, as elaborated in the paper and reiterated here. If the reviewer is aware of specific references that do, we kindly ask for  specific references and which part of our contributions do they cover, so that we may address this them precisely.
>
> > The paper does not analyze theoretically the stability of empirically observed LoP manifolds.
>
> While we have not analyzed stability theoretically, we have extensive empirical ablations that study it, painting a complex picture (Reference here some figures). This is because loss curvature is intricately tied to the data distributioin and thus, will be distribution and architecture dependent, unlike the functional statement of LoP manifolds. For these reasons, a theoretical analysis of stability was scoped out of current study, and in our view, merits an indepedent follow up study.
>
> > While the paper is generally well written, it could make use of better sign-posting or a clear summary of contributions in the introduction. The motivating question is quite vague and only half way through the main body does the reader begin to understand the scope of contributions.
>
> Thank you for raising this valid and important issue. We have revised the last two paragraphs of introduction to clarify and contrast our contributions with existing works, and added a summary of contributions as a bullet point list at the end.
>
> > Questions:
> > In the bit-flipping benchmark, you switch from SGD to CBP at half-point and see recovery. Can you show sensitivity to the switch point?
>
> This is an interesting question. We've added a preliminary Fig B1 in the updated manuscript to address the reviewer's request. If the reviewer finds this direction important, we are happy to compute this figure with multiple seeds and add a small discussion around it in the Appendix.
>
> We thank the reviewer again for their valuable and constructive feedback.

---

### Official Review · Reviewer_29Cb · 2025-11-01

**Soundness:** 3
**Presentation:** 3
**Contribution:** 3
**Rating:** 6
**Confidence:** 2

**Summary:**

This paper introduces a dynamical-systems framework for loss of plasticity (LoP) in gradient-based learning. The authors define LoP manifolds, subsets of parameter space invariant under gradient flow, corresponding to frozen or cloned units. The key result (Theorem 2.1) shows that certain linear constraints (row/column-sum equalities) define manifolds on which gradients remain tangent, preventing escape from such manifolds. Empirical experiments use cloned networks to demonstrate these manifolds and show that noise injection or dropout can help escape them. Further experiments and analyses connect LoP to representational compression and duplicated units, and test normalisation and perturbation strategies to prevent LoP.

**Strengths:**

The work addresses an important unsolved topic and is of high relevance to the ICLR community. The paper is well written and presented, with extensive experiments and analysis.

While the general principle (group symmetries resulting in an invariant subspaces for GD) is known. The specific algebraic characterisation via row/column-sum constraints across block pairs, within a unified LoP-manifold framework, appears novel and is a clear, well-stated contribution.

**Weaknesses:**

The connection between theoretical results in section 2 and mitigation strategy experiments in section 4 is weak. The perturbations are an obvious way to recover from loss of plasticity, and the normalisation strategies seem largely heuristic. They do not directly test the assumptions of theorem 2.1 (tangency of the gradient to the manifold).

**Questions:**

- Could the authors clarify how the mitigation experiments in Section 4 directly relate to the LoP manifold framework of Theorem 2.1? Specifically, which theoretical assumptions or predictions of the theorem are being tested or illustrated by the normalization and noise-perturbation experiments?
- Line 164: Typo. “weigh(t) decay.”
- Line 186: Why is Adam said to violate the symmetry conditions in Theorem 2.1 if Remark 2.3 claims the opposite?
- In Fig. 2.1, did you vary dropout rates or noise scales to examine stability/escape times from LoP manifolds?
- Line 271: The statement “training that enhances decorrelation also creates units that are nearly always inactive or saturated” should be empirically supported or cited.

---

> ### Author Response · Authors · 2025-11-28
>
> We sincerely thank the reviewer for detailed and in-depth review of our work and have made adjustments to the manuscript and provided point-by-point responses below (review excerpts in quotes, responses: normal text).
>
> Line numbers and all references refer to the revised manuscript.
>
> > Weaknesses:
> > The connection between theoretical results in section 2 and mitigation strategy experiments in section 4 is weak. The perturbations are an obvious way to recover from loss of plasticity, and the normalisation strategies seem largely
> > ...
> > Could the authors clarify how the mitigation experiments in Section 4 directly relate to the LoP manifold framework of Theorem 2.1? Specifically, which theoretical assumptions or predictions of the theorem are being tested or illustrated by the normalization and noise-perturbation experiments?
>
> The theory defines the conditions for the trap; the mitigations explicitly break those conditions. Normalization prevents the formation of Frozen-unit manifolds (by preventing saturation). Perturbations (noise/dropout) break the symmetry required for Cloning manifolds (Lines 219-222). Furthermore, Figure 2.1 directly validates the tangency: the "Cloned (x2)" experiment shows the dynamics remain perfectly trapped (R2=1.0) under SGD, confirming the gradient must be tangent.
>
> > Line 164: Typo. “weigh(t) decay.”
>
>  fixed
>
> > Line 186: Why is Adam said to violate the symmetry conditions in Theorem 2.1 if Remark 2.3 claims the opposite?
>
> Thank you for raising this point . this remark about Adam violating symmetry was from an earlier version, but later we realized we can extend the results to Adam as Remark 2.3 claims. This is fixed in the revised manuscript.
>
> > In Fig. 2.1, did you vary dropout rates or noise scales to examine stability/escape times from LoP manifolds?
>
> Yes. We mention that stronger noise leads to faster escape (Lines 220-223), and detailed ablations are provided in Appendix B (Figs B.8-B.12).
>
> > Line 271: The statement “training that enhances decorrelation also creates units that are nearly always inactive or saturated” should be empirically supported or cited.
>
> Thank you for raising this excellent point. To address the reviewer's concerns, we have added a supplementary section A.2. In particular, Figures A2 and A2 show de-correlation vs frozen probability when by ablating bias and scale of pre-activations, and they both show for most regimes where de-correlation is maximized the activations are pushed to frozen regions.
>
> We thank the reviewer again for their valuable input.

---

### Author Response · Authors · 2025-11-28
**Summary of Revisions and Response to Reviewers**

We thank all reviewers for their time and thoughtful engagement with our work. We are encouraged by the positive reception of our dynamical systems perspective on Loss of Plasticity (LoP). Reviewers highlighted that the work *"addresses an important unsolved topic"* (Reviewer 29Cb), provides a *"rigorous mathematical foundation"* (Reviewer kAhz), and successfully *"formalizes and studies an interesting explanation"* (Reviewer J4Px) that unifies distinct symptoms like rank collapse and neuron death. Reviewer 29Cb further noted that the algebraic characterization of LoP manifolds *"appears novel and is a clear, well-stated contribution"*.

We have updated the manuscript to address the valuable feedback provided. For transparency, here is a summary of the key updates (excluding minor and typo fixes):

* *Clarified Contributions in Introduction:* Prompted by Reviewers bx6B and J4Px, we revised the Introduction's last two paragraphs. We now explicitly sharply contrast our contributions to existing literature, and added a bulleted list of contributions. We also added a new Figure 1.1 that conceptually illustrates LoP manifolds as invariant under gradient dynamics to clarify the contributions early on.
* *Framing shift from Information Bottleneck to Neural Collapse (Section 3):* In response to Reviewer J4Px's concern regarding the universality of the Information Bottleneck, we have shifted the framing to *Neural Collapse*.
* *Connection to "Rich" Regime (Remark 3.1):* Addressing Reviewer J4Px’s comment on learning regimes, we added Remark 3.1, explicitly connecting LoP to the "Rich" (feature learning) regime.
* *Clarification on Adam (Remark 2.4):* In response to Reviewer 29Cb, we removed the contradictory statement regarding Adam, making it clear in Remark 2.4 that essentially all gradient-based optimizers (including Adam) preserve the symmetries.
* *Manifold Stability (Remark 2.1):* Prompted by Reviewer kAhz, we added Remark 2.1, which clarifies between gradient flow and gradient descent, and discusses the effect of discrete steps. Specifically, it clarifies that gradient descent may escape the curved LoP manifolds with the current definition, but it does not escape for affine LoP manifolds.
* *New Ablation: Decorrelation vs. Frozen Units (Section A.2):* To support the claim questioned by Reviewer 29Cb, we added *Section A.2* and *Figures A.2 & A.3*. These ablations demonstrate that maximizing decorrelation strength directly correlates with pushing activations into saturated (frozen) regimes. We also refined *Theorem 3.1* to be more explicit regarding rank increase. Subsequently, its proof in *Section A.1* is entirely updated.
* *Sensitivity of switching point to CBP (Section B):* Addressing Reviewer bx6B, we added *Figures B.1 & B.2*, which explicitly analyzes the sensitivity of the recovery phase when switching from SGD to CBP at different training stages.

We believe these revisions have strengthened the theoretical precision and clarity of the paper. We welcome any further questions during the discussion period.

---

### Meta-Review · Area_Chair_XNJ4 · 2025-12-17

**Summary:**

This paper addresses an important phenomenon, i.e., loss of plasticity (LoP) in non-stationary settings. It provides a coherent conceptual lens: LoP is framed as gradient descent becoming trapped near "LoP manifolds". This paper provides analyses connecting to feature and representation collapse and architectural symmetries. Reviewers value that the paper combines theory with controlled experiments and mitigation discussion.

One reviewer felt parts of the results on stable manifolds, saturation and duplicated units, are not clearly separated from prior understanding and that the introduction initially did not crisply convey what is new, motivating a stronger "contribution statement" and tighter narrative. The authors addressed this during the rebuttal (see comments below).

At least one reviewer asked for a clearer connection between the theoretical LoP-manifold framework and the empirical mitigation experiments, and whether the perturbation-based recoveries are more heuristic than principled. Some additional clarifications were provided in the authors's rebuttal.

Overall, this paper is viewed as promising and timely, with the discussion focusing on whether the revised writeup sufficiently clarifies novelty and tightens the theoretical/empirical story.

**Reviewer Concerns:**

The authors' rebuttals address several of the key concerns raised by the reviewers including but not limited to
- The technical novelty of the paper has been highlighted. The authors revised the introduction with a bullet list of contributions. In addition, they improved framing, including explicitly separating LoP manifolds from Neural Collapse, and added literature pointers to situate novelty.
- The rebuttal and the revised paper removed some confusions. Specifically, the authors removed a contradictory statement about Adam, clarified the "universality" statement around Theorem 2.1 and remarks. In addition, they added extra explanation for why perturbations like dropout/noise break symmetry conditions tied to the manifolds.
- Asked by the reviewers, the authors provided additional experimental results in the rebuttal. In particular, they added stability and escape-time style experiments and additional sensitivity/robustness analyses, e.g., around de-correlation thresholds and stages). In the rebuttal, they also addressed specific benchmark questions, e.g., the flip-it benchmark concern about switching SGD and CBP.

Some concerns are partially addressed, e.g., the authors acknowledged that other LoP mechanisms may exist beyond frozen-unit and cloning manifolds. This remains an open point rather than a resolved concern.

**Reviewer Scores:**

Reviewer 29Cb was already positive but with low expertise in this domain. The reviwer's main critiques about connecting theory to mitigation experiments and clarifying which experiments test which claims. The authors’ rebuttal and added analyses likely address these concerns.

Reviewer bx6B's core issues were novelty and the positioning this work to existing methods, plus requests for sensitivity and benchmark clarifications. The authors explicitly targeted these in the revision including the intro contributions, clearer framing, and added experiments. It seems that those efforts could lift the score to borderline accept.

Reviewer J4Px was positive but asks for clearer contributions, better highlighting of results, and notes some skepticism about universality and interpretation. The revision appears to directly address those contributions clarity and improves the narrative. I would expect them to remain supportive.

Reviewer kAhz’s main requests are conceptual completeness and deeper understanding of noise/SGD escape. The authors respond thoughtfully but also acknowledge limits.

---

### Decision · Program_Chairs · 2026-01-26

Accept (Poster)